   Select

# Thermal conformal blocks

### Yan Gobeil[1]⋆, Alexander Maloney [1]†, Gim Seng Ng [2,3]‡ Jie-qiang Wu [4]○

**1** Department of Physics, McGill University, Montréal, QC, Canada
**2** School of Mathematics, Trinity College Dublin, Dublin 2, Dublin, Ireland
**3** Hamilton Mathematical Institute, Trinity College Dublin, Dublin 2, Ireland
**4** Center for Theoretical Physics, Massachusetts Institute of Technology,
Cambridge, MA 02139, USA

⋆ yan.gobeil@mail.mcgill.ca, † maloney@physics.mcgill.ca, ‡ gng@math.tcd.ie, ○ jieqiang@mit.edu

## Abstract

We study conformal blocks for thermal one-point-functions on the sphere in conformal field theories of general dimension. These thermal conformal blocks satisfy second-order Casimir differential equations and have integral representations related to AdS Witten diagrams. We give an analytic formula for the scalar conformal block in terms of generalized hypergeometric functions. As an application, we deduce an asymptotic formula for the three-point coefficients of primary operators in the limit where two of the operators are heavy.

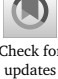
# 1 Introduction

The recent revival of the conformal bootstrap program (see [1–5] for reviews) has led to impressive advances in our understanding of conformal field theories (CFTs). The main strategy of this program is to impose the constraints of unitarity and conformal invariance directly on the theory, without relying on a traditional perturbative expansion. So far most work has focused on the constraints coming from crossing symmetry of flat-space four point functions, which is a consequence of the associativity of the operator product expansion (OPE). The constraints of conformal invariance on other observables, such as correlation functions in other backgrounds or at finite temperature, are less well understood. A notable exception is in two dimensions, where modular invariance relates the high temperature behaviour of the theory to the low temperature behaviour. But the constraints coming from the consistency of higher dimensional CFTs at finite temperature have not received as much attention.[1]

In this work, we aim to provide a first step towards a bootstrap program for thermal correlators in higher dimensional CFT. In the case of the four point functions, the essential ingredients in the bootstrap program are the conformal blocks, which describe the contributions of an entire representation of conformal symmetry to a four point function. We are therefore interested in exploring the structure of finite temperature conformal blocks, which describe the contribution of a particular representation (or set of representations) of conformal symmetry to a given thermal correlation function. We will study conformal field theory in $d > 2$ space-time dimensions on $\mathbb{R}_{\text{time}} \times S^{d-1}$ at finite temperature; we will sometimes specialize to $d = 3$ for the sake of definiteness. We will focus on the computation of the conformal blocks

---

[1]See, however, [6–16]. For more recent progress, see [17].

for one-point functions of scalar operators at finite temperature, which is the first non-trivial case.[2] We will obtain explicit expressions for these blocks when the internal primary operator is also a scalar.

We begin in Sec. 2 by describing the Casimir differential equations satisfied by these thermal conformal blocks. These are second order differential equations, which are obtained by studying the insertion of the conformal Casimir operator into the thermal trace. In Sec. 2 we will present explicit formulas for $d = 3$. In Appendix B we will discuss the generalization to higher dimensions, which is straightforward, although the explicit expressions are more complicated. This thermal Casimir method can be viewed as the generalization of the Casimir method of [23] to thermal one-point functions, or as a higher-dimensional generalization of the two dimensional case presented in [21]. One notable feature of this differential equation is that it involves not just derivatives with respect to temperature, but also with respect to angular potentials for the spatial sphere $S^{d-1}$. Thus, in seeking to find explicit solutions of the Casimir equation it is necessary to consider the block with all possible thermodynamic potentials turned on.

In Sec. 3 we will describe an integral representation of the thermal one-point block. Although we are not assuming the existence of a holographic dual, this integral representation can be interpreted in terms of a Witten diagram in AdS. This is quite similar to the AdS geodesic Witten diagram representation of the flat space conformal blocks [24–29].[3] We will first give a heuristic motivation for our integral representation by considering the decomposition of the full Witten diagram calculation of CFT thermal one-point functions into conformal blocks. Then in Sec. 3.2 we will give a more constructive proof of our AdS-integral representation using the shadow formalism of [33]. In Appendix E we will verify that our integral expression solves the thermal Casimir equation with the correct boundary conditions. Remarkably, it turns out that this integral expression can be evaluated for the scalar conformal block in the absence of angular potentials. This will give an explicit analytic expression for the thermal block in terms of the generalized hypergeometric function $_3F_2$. The integral expression will also allow us to easily discuss various limits of the block.

We will conclude in Sec. 4 with an important physical application, which is an asymptotic formula for the OPE coefficients of primary operators. At high temperature, the one-point function of an operator $\phi$ of dimension $\Delta_\phi$ will take the form[4]

$$\langle \phi \rangle_\beta \approx \alpha_\phi \beta^{-\Delta_\phi} + \dots, \tag{1}$$

where $\alpha_\phi$ is a constant which depends on the theory and the choice of operator $\phi$. The inverse Laplace transform of this equation then gives an expression for this microcanonical expectation value of $\phi$:

$$\overline{\langle \mathcal{O}| \phi |\mathcal{O}\rangle}\Big|_{\Delta_\mathcal{O}} \approx \alpha_\phi \left(\frac{\Delta_\mathcal{O}}{(d-1)\tilde{c}}\right)^{\frac{\Delta_\phi}{d}}. \tag{2}$$

This is the *average* value of the expectation value of $\phi$ in a heavy state $|\mathcal{O}\rangle$, averaged over all states $|\mathcal{O}\rangle$ of energy $\Delta_\mathcal{O}$, and $\tilde{c}$ is a theory-dependent constant related to the asymptotic density

---

[2]Although we are not aware of a detailed study of thermal conformal blocks for $n$-point functions in dimension $d > 2$, there is a considerable literature (see e.g. [18–22]) for two dimensional CFTs.

[3]We note that, for flat space four-point blocks, there are several other interesting representations of the conformal blocks that should exist for thermal blocks as well. For example, one representation uses the analytical properties of the four-point block to obtain a recursion relation [30,31]. Another involves dimensional reduction of the conformal block to lower-dimensional conformal blocks [32]. It would be interesting to generalize these representations to thermal conformal blocks. We thank David Poland and Eric Perlmutter for discussions related to this.

[4]It is possible for the coefficient $\alpha_\phi$ to vanish, in which case one might have to worry about subleading terms. This occurs in $d = 2$, for example, where $\alpha_\phi = 0$ by conformal invariance but there are non-vanishing subleading corrections (which depend on the sphere radius) that vanish exponentially as $\beta \to 0$ (as in [20]). In $d > 2$ we generically expect $\alpha_\phi \neq 0$ unless it is set to zero by a symmetry. So we will assume $\alpha_\phi \neq 0$ in what follows.

of states. Since $\langle \mathcal{O} | \phi | \mathcal{O} \rangle \sim C_{\mathcal{O}\phi\mathcal{O}}$ is an OPE coefficient, this can be viewed as an asymptotic formula for the average value of the light-heavy-heavy three-point coefficients.[5] Although we will not focus on it here, this asymptotic formula plays a role in the Eigenstate Thermalization Hypothesis (ETH), as discussed in [34]. A priori, equation (2) includes an average over all operators $\mathcal{O}$, not just primaries. However, now that we have explicit expressions for the one-point thermal blocks we can obtain the analog of equation (2) where we average only over primary operators $\mathcal{O}$, instead of averaging over all operators. Remarkably, we will see in Sec. 4 that the result is still exactly given by equation (2).

Before moving on to the main part of the paper, let us first introduce some preliminary definitions and state carefully our main result: an explicit formula for the scalar one-point block.

## 1.1 Conformal blocks: definitions and conventions

We first need to introduce our conventions. For flat Euclidean $\mathbb{R}^d$ we will use either cartesian coordinates $x^\mu$, $\mu = 1, \ldots d$, or spherical coordinates $(r, \Omega_j)$, $j = 1, \ldots d-1$, with:

$$ds^2_{\mathbb{R}^d} = dx^\mu dx_\mu = dr^2 + r^2 d\Omega^2_{d-1}. \tag{3}$$

Letting $r = e^\tau$, this metric is conformally related to the cylinder $\mathbb{R} \times S^{d-1}$

$$ds^2_{\mathbb{R}^d} = dx^\mu dx_\mu = e^{2\tau}(d\tau^2 + d\Omega^2_{d-1}) = e^{2\tau} ds^2_{\mathbb{R} \times S^{d-1}} \tag{4}$$

in the usual radial quantization map. A scalar primary operator $\phi_{\mathbb{R}^d}$ on the plane of definite conformal weight $\Delta_\phi$ will transform to a cylinder operator $\phi_{\mathrm{cyl}}$ as

$$\phi_{\mathbb{R}^d}(x) = e^{-\tau \Delta_\phi} \phi_{\mathrm{cyl}}(\tau, \Omega). \tag{5}$$

We shall reserve the unsubscripted $\phi$ to be the operator on flat space $\phi_{\mathbb{R}^d}$ while the cylinder operator $\phi_{\mathrm{cyl}}$ will carry the subscript 'cyl'.

The thermal one-point function of a scalar primary operator $\phi_{\mathrm{cyl}}$ is defined as its (unnormalized) thermal expectation value:

$$
\begin{aligned}
\left\langle \phi_{\mathrm{cyl}}(\tau, \Omega) \right\rangle_\beta &\equiv \mathrm{Tr}\left[ \phi_{\mathrm{cyl}}(\tau, \Omega) e^{-\beta D} \right] = \sum_{i,j} e^{-\beta \Delta_i} \langle i | \phi_{\mathrm{cyl}} | j \rangle_{\mathbb{R} \times S^{d-1}} (B^{-1})^{ji} \\
&= \sum_{i,j} e^{-\beta \Delta_i} e^{\tau \Delta_\phi} \langle i | \phi(x) | j \rangle_{\mathbb{R}^d} (B^{-1})^{ji} = \sum_{i,j} e^{-\beta \Delta_\mathcal{O}} C_{i\phi j} (B^{-1})^{ji}.
\end{aligned} \tag{6}
$$

Here, $D$ is the dilatation operator[6] while the trace is over the Hilbert space of the CFT quantized on $\mathbb{R} \times S^{d-1}$ (where the radius of the sphere is one). In the second line, we relate the thermal one-point function to objects on $\mathbb{R}^d$. In particular the OPE coefficient $C_{i\phi j}$ is defined through three-point function on $\mathbb{R}^d$: $\langle i | \phi(x) | j \rangle_{\mathbb{R}^d} = C_{i\phi j} r^{-\Delta_\phi}$. We note that $\left\langle \phi_{\mathrm{cyl}}(\tau, \Omega) \right\rangle_\beta$ depends on $\beta$, but is independent of $(\tau, \Omega)$ by translation invariance. In this expression the sum over $i$ includes contributions from all states, and $B_{ij} = \langle i | j \rangle$ will not necessarily be taken to be diagonal.

We now organize the sum in terms of conformal primaries:

$$\left\langle \phi_{\mathrm{cyl}} \right\rangle_\beta = \sum_{\text{primary } \mathcal{O}} C_{\mathcal{O}\phi\mathcal{O}} \mathcal{F}_{\Delta_\phi; \Delta_\mathcal{O}, \ell_\mathcal{O}}(q), \tag{7}$$

---

[5]It is important that we are taking $d > 2$ here; the analagous expressions in two dimensions are very different [20].

[6]See Appendix A for more details on our conventions for the conformal group and its representations.

where $q \equiv e^{-\beta}$. The functions $\mathcal{F}_{\Delta_\phi;\Delta_\mathcal{O},\ell_\mathcal{O}}(q)$ are the thermal one-point conformal blocks, and encode the contributions from the conformal descendants of the primary operator $\mathcal{O}$, which we will take to have dimension and spin $(\Delta_\mathcal{O}, \ell_\mathcal{O})$. They are completely fixed by conformal symmetry, and depend *only* on $\Delta_\phi, \Delta_\mathcal{O}, \ell_\mathcal{O}$ and $q$. Comparing Eq. (6) and Eq. (7), we have the expansion

$$\mathcal{F}_{\Delta_\phi;\Delta_\mathcal{O},\ell_\mathcal{O}}(q) = q^{\Delta_\mathcal{O}} \sum_{k=0}^{\infty} q^k \sum_{|m|=|n|=k} \frac{\langle \mathcal{O}, \vec{m} | \phi(1) | \mathcal{O}, \vec{n} \rangle}{C_{\mathcal{O}\phi\mathcal{O}}} (B^{-1})^{\vec{m},\vec{n}}, \tag{8}$$

where $|\mathcal{O}, \vec{n}\rangle$ are the descendants defined in Eq. (93).

When $\ell_\mathcal{O} \neq 0$ one needs to carefully distinguish the different tensor structures of $\mathcal{O}$ (whose tensor indices have been suppressed). There is a different block for each tensor structure in the three-point function $\langle \mathcal{O}\phi\mathcal{O} \rangle$, and the results can become quite complicated. We shall therefore focus on the case where $\ell_\mathcal{O} = 0$, and thus consider the scalar conformal block

$$\mathcal{F}_{\Delta_\phi;\Delta_\mathcal{O}}(q) \equiv \mathcal{F}_{\Delta_\phi;\Delta_\mathcal{O},\ell_\mathcal{O}=0}(q). \tag{9}$$

## 1.2 Explicit form of conformal blocks

In principle, using Eq. (8), one can compute by brute force the coefficients of the $q$ expansion of the thermal block. An algorithm is presented in Appendix C to do so, which can be implemented in Mathematica. However, as we shall see in Sec. 2, these thermal blocks (with appropriate angular momentum potentials turned on) satisfy a Casimir differential equation similar to the flat space conformal blocks. As such, one hopes that by solving the differential equation, a closed-form explicit formula can be obtained. We will, however, not attempt to solve these equations directly. Instead, motivated by recent work on the AdS representation of the conformal blocks [21, 26–29], in Sec. 3, we will obtain an AdS-integral representation for the thermal conformal blocks. For the case of zero angular potentials, this AdS-integral representation yields an explicit closed-form expression for the conformal block:

$$\mathcal{F}_{\Delta_\phi;\Delta_\mathcal{O}}(q) = q^{\Delta_\mathcal{O}}(1-q)^{-2\Delta_\mathcal{O}}$$
$$\times {}_3F_2\left(-\frac{d}{2} + \Delta_\mathcal{O} + \frac{1}{2}, \Delta_\mathcal{O} - \frac{\Delta_\phi}{2}, -\frac{d}{2} + \frac{\Delta_\phi}{2} + \Delta_\mathcal{O}; \Delta_\mathcal{O}, -d + 2\Delta_\mathcal{O} + 1; -\frac{4q}{(q-1)^2}\right). \tag{10}$$

This expression looks very similar to the so-called diagonal limit of the flat space four-point block in [2, 35]. Although we have not done so in this paper, it will be interesting to study the diagonal limit of our Casimir equation and derive Eq. (10) in this manner.[7]

## 2 Thermal Casimir method

In this section we derive a Casimir differential equation for thermal conformal blocks. We will use a generalization of the technique of [21], who studied two dimensional thermal blocks. In this section we will focus on $d = 3$, and write down the Casimir equation completely explicitly. In Appendix B we will discuss the differential equation in general dimension.

Our conventions for the conformal group and its representations are summarized in Appendix A.

---

[7] As a final remark, we note recent interesting work relating flat space conformal blocks to wave functions of integrable systems [36–42], where the conformal Casimirs are mapped to the Hamiltonian and higher conserved charges. It will be interesting to explore these connections in the context of thermal conformal blocks. We thank Samson Shatashvili for discussions related to this.

## 2.1 Casimir operator in 3d

For $d = 3$, the rotational subgroup (generated by the $M_{\mu\nu}$) of the conformal group is $SO(3)$. We define

$$J_3 \equiv -iM_{12}, \quad J_\pm \equiv iM_{23} \pm M_{13}, \tag{11}$$

so that

$$[J_3, J_\pm] = \pm J_\pm, \quad [J_+, J_-] = 2J_3. \tag{12}$$

It will be useful to consider the following coordinate system for $\mathbb{R}^3$:

$$x^\pm \equiv x^{(1)} \pm ix^{(2)}, \quad ds^2_{\mathbb{R}^3} = dx^+ dx^- + (dx^{(3)})^2, \tag{13}$$

with

$$r^2 = x^+ x^- + (x^{(3)})^2, \quad \partial_1 = \partial_+ + \partial_-, \quad \partial_2 = i(\partial_+ - \partial_-), \quad \partial_\pm = \frac{\partial_1 \mp i\partial_2}{2}. \tag{14}$$

We will sometimes use the notation $x^{(\mu)} \equiv x^\mu$ to avoid confusion with various powers of $x^\mu$. We will also use spherical coordinates on $\mathbb{R}^3$:

$$x^{(1)} = rc_\varphi s_\theta, \quad x^{(2)} = rs_\varphi s_\theta, \quad x^{(3)} = rc_\theta, \tag{15}$$

where $c_x \equiv \cos x$ and $s_x \equiv \sin x$.

To work with these new coordinates, we define the operators

$$P_\pm \equiv P_1 \pm iP_2, \quad K_\pm \equiv K_1 \pm iK_2. \tag{16}$$

With these definitions, in radial quantization, we have $P_\pm^\dagger = K_\mp$ and $P_0^\dagger = K_0$. Using our new notation, the action of the conformal generators on scalar primaries (94) becomes

$$
\begin{aligned}
{[D, \phi(x)]} &= (\Delta_\phi + x^\mu \partial_\mu)\phi(x) \equiv \mathcal{D}\phi, \\
{[P_\pm, \phi(x)]} &= (2\partial_\mp)\phi \equiv \mathcal{P}_\pm \phi, \\
{[K_\pm, \phi]} &= \left(2x_\pm \Delta_\phi + 2x_\pm(x \cdot \partial) - x^2(2\partial_\mp)\right)\phi \equiv \mathcal{K}_\pm \phi, \\
{[J_\pm, \phi(x)]} &= (\pm x_3(2\partial_\mp) \mp x_\pm \partial_3)\phi \equiv \mathcal{J}_\pm \phi, \\
{[J_3, \phi]} &= (x^- \partial_- - x^+ \partial_+)\phi \equiv \mathcal{J}_3 \phi,
\end{aligned}
\tag{17}
$$

where curly letters denote spatial derivatives on operators. The quadratic Casimirs are:

$$
\begin{aligned}
J^2 &\equiv -\frac{1}{2}M_{\mu\nu}M^{\mu\nu} = J_3(J_3 + 1) + J_- J_+, \\
C &\equiv D(D-3) + J^2 - P_0 K_0 - \frac{1}{2}(P_+ K_- + P_- K_+).
\end{aligned}
\tag{18}
$$

$J^2$ is the Casimir of the $SO(3)$ algebra and $C$ is the Casimir of the conformal algebra.

## 2.2 General structure

Let us focus on the case of scalar internal and external operators. To study the contribution of a single conformal family (say generated by $\mathcal{O}$) to the thermal expectation value, we need to insert the projection operator

$$P_\mathcal{O} = \sum_{i,j=\mathcal{O}, P\mathcal{O}, P^2\mathcal{O}, \dots} |i\rangle (B^{-1})_{ij} \langle j| \tag{19}$$

into the thermal trace.

It turns out that we will need to turn on both temperature and angular momentum potentials in the trace in order to use the thermal Casimir method. The object that we are interested in is then

$$\mathcal{F}_{\Delta_\phi,\Delta_\mathcal{O}}(q,y;x) = C_{\mathcal{O}\phi\mathcal{O}}^{-1} \text{Tr}\left[P_\mathcal{O}\phi_{\text{cyl}}q^D y^{J_3}\right] = C_{\mathcal{O}\phi\mathcal{O}}^{-1} r^{\Delta_\phi} \text{Tr}\left[P_\mathcal{O}\phi(x)q^D y^{J_3}\right]. \qquad (20)$$

Here $y$ is the potential for the angular momentum $J_3$. When $y = 1$, i.e. with no angular potential, the conformal block $\mathcal{F}$ has no $x$-dependence. But when $y \neq 1$, $\mathcal{F}$ will depend non-trivially on $x$.

Let us now study this $x$-dependence. To begin, we insert $D$ into the trace, which we denote by $\text{Tr}[P_\mathcal{O}...] = \text{Tr}_\mathcal{O}[...]$:

$$
\begin{aligned}
\text{Tr}_\mathcal{O}\left[D\phi(x)q^D y^{J_3}\right] &= \text{Tr}_\mathcal{O}\left[[D,\phi(x)]q^D y^{J_3}\right] + \text{Tr}_\mathcal{O}\left[\phi(x)q^D y^{J_3}D\right] \\
&= \mathcal{D}\text{Tr}_\mathcal{O}\left[\phi(x)q^D y^{J_3}\right] + \text{Tr}_\mathcal{O}\left[D\phi(x)q^D y^{J_3}\right] \\
\implies 0 &= (\Delta_\phi + x^\mu \partial_\mu)\mathcal{F}.
\end{aligned}
\qquad (21)
$$

Here we have used Eq. (17) to move $D$ to the right, used the fact that $D$ and $J_3$ commute, and used the cyclicity of the trace. The result is

$$\mathcal{F}_{\Delta_\phi,\Delta_\mathcal{O}}(q,y;x) = \text{Function}\left(q,y;\frac{x^+}{x^{(3)}},\frac{x^-}{x^{(3)}}\right). \qquad (22)$$

Similarly, inserting and moving $J_3$ through implies that $\mathcal{F}$ is a function of $x^- x^+$ and $x^{(3)}$. Combining these two facts, we conclude that we can write the conformal block in the form

$$\mathcal{F}_{\Delta_\phi,\Delta_\mathcal{O}}(q,y;x) = \text{Function}\left(q,y;\frac{x^+ x^-}{r^2}\right) = \text{Function}(q,y;s), \quad s \equiv s_\theta^2. \qquad (23)$$

## 2.3 Casimir differential equation

Let us begin with the following equations:

$$
\begin{aligned}
P_\pm q^D y^{J_3} &= q^{D-1}y^{J_3 \mp 1}P_\pm, & P_3 q^D y^{J_3} &= q^{D-1}y^{J_3}P_3, \\
K_\pm q^D y^{J_3} &= q^{D+1}y^{J_3 \mp 1}K_\pm, & K_3 q^D y^{J_3} &= q^{D+1}y^{J_3}K_3, \\
J_\pm q^D y^{J_3} &= q^D y^{J_3 \mp 1}J_\pm.
\end{aligned}
\qquad (24)
$$

These are true as operator statements, as can be seen by expanding the exponentials, but one can also derive them by acting on an orthogonal basis for the descendants where the states are labelled by their $D$, $J_3$ and $J^2$ eigenvalues. This is straightforward but not essential for our calculation, so we will not discuss this here.

The thermal Casimir trick is to insert the Casimir operator defined above into the trace (20). We can then use Ward identities to convert each of the terms into derivatives acting on $\mathcal{F}$. On the other hand, the Casimir operator has value $\Delta_\mathcal{O}(\Delta_\mathcal{O} - 3)$ for each state in a scalar conformal family. This leads to a Casimir equation of the form

$$(x^2)^{\Delta_\phi/2} C_{\mathcal{O}\phi\mathcal{O}}^{-1} \text{Tr}_\mathcal{O}\left[C\phi(x)q^D y^{J_3}\right] = \Delta_\mathcal{O}(\Delta_\mathcal{O} - 3)\mathcal{F} = \mathcal{CF}, \qquad (25)$$

where $\mathcal{C}$ is the differential operator associated with the Casimir operator.

We now just need to compute the differential operator $\mathcal{C}$. First, note that inserting $D$ in the trace $\text{Tr}_\mathcal{O}\left[\phi(x)q^D y^{J_3}\right]$ is equivalent to acting on the trace with $q\partial_q$. This allows us to convert the first term of the Casimir (see Eq. (18)) into derivatives. Similarly, inserting $J_3$ is the same as acting with $y\partial_y$. This is why we need to include an angular potential in the trace – otherwise, we would be unable to evaluate $J_3$ in the Casimir. For the other operators, we

use Eq. (17), Eq. (24), and the conformal algebra to bring the operators to the right of $\phi(x)$. Cyclicity of the trace allows us to combine some terms and convert the quantum operators to differential operators. For example,

$$
\begin{aligned}
\text{Tr}_{\mathcal{O}}\left[J_+\phi(x)q^D y^{J_3}\right] &= \text{Tr}_{\mathcal{O}}\left[[J_+,\phi(x)]q^D y^{J_3}\right] + \text{Tr}_{\mathcal{O}}\left[\phi(x)J_+q^D y^{J_3}\right] \\
&= \mathcal{J}_+\text{Tr}_{\mathcal{O}}\left[\phi(x)q^D y^{J_3}\right] + \text{Tr}_{\mathcal{O}}\left[\phi(x)q^D y^{J_3-1}J_+\right] \\
&= \mathcal{J}_+\text{Tr}_{\mathcal{O}}\left[\phi(x)q^D y^{J_3}\right] + y^{-1}\text{Tr}_{\mathcal{O}}\left[J_+\phi(x)q^D y^{J_3}\right], \\
\Rightarrow \text{Tr}_{\mathcal{O}}\left[J_+\phi(x)q^D y^{J_3}\right] &= \frac{1}{1-y^{-1}}\mathcal{J}_+\text{Tr}_{\mathcal{O}}\left[\phi(x)q^D y^{J_3}\right].
\end{aligned} \tag{26}
$$

Eventually, all of the operators appearing in the Casimir are converted into differential operators on $(q,u,s)$ and we obtain the Casimir differential equation:

$$
\begin{aligned}
0 = {}& q^2\partial_q^2 f + u(u+4)\partial_u^2 f + 2q\left(\Delta_{\mathcal{O}} + \frac{q(2q-u-2)}{q^2-q(u+2)+1} + \frac{q}{q-1} - 1\right)\partial_q f \\
&+ \frac{2(q(q(u+3)-u(2u+9)-6)+u+3)}{q^2-q(u+2)+1}\partial_u f \\
&+ \frac{2\Delta_{\mathcal{O}}q\left(3q^2-2q(u+3)+u+3\right)}{(q-1)^3-(q-1)qu}f \\
&- u^{-1}\left[2(3s-2)\partial_s + 4s(s-1)\partial_s^2\right]f\,\frac{q}{(1-q)^2} \\
&+ \left[\Delta_\phi(1-\Delta_\phi)+\Delta_\phi^2 s + \left[-2(1+2\Delta_\phi)+4(\Delta_\phi+1)s\right]s\partial_s + 4s^2(s-1)\partial_s^2\right]f \\
&+ \left[\frac{q(q(q(u+2)-4)+u+2)}{2\left(q^2-q(u+2)+1\right)^2}\right] \tag{27} \\
&\times \left[2\Delta_\phi - \Delta_\phi^2 s + 4\left[(\Delta_\phi+2)s-(\Delta_\phi+1)s^2-1\right]\partial_s - 4s(1-s)^2\partial_s^2\right]f.
\end{aligned}
$$

$$\tag{28}$$

Here $u \equiv y + y^{-1} - 2$ (we pick the root $y = (u/2)+1+\sqrt{(u/2)((u/2)+2)}$) and we have defined $f$ to be

$$
f_{\Delta_\phi,\Delta_{\mathcal{O}}}(q,y;s) \equiv q^{-\Delta_{\mathcal{O}}}\mathcal{F}_{\Delta_\phi,\Delta_{\mathcal{O}}}(q,y;s) = 1 + \mathcal{O}(q^1). \tag{29}
$$

Note that the differential equation is a 2nd-order differential equations in *three* variables (i.e. $q,u$ and $s$).

Now that we have the differential equation, let us discuss the boundary conditions the solution must obey. We are looking for solutions which have appropriate behavior at small $q$: as $q \to 0$, the solution must approach 1. We also want the $u \to 0$ limit to give the $q$ expansion of the zero-angular-rotation thermal block, which is independent of $s$. These conditions are sufficient to fix a unique solution. More precisely, we can imagine expanding the block in a power series:

$$
f(q,u,s) = \sum_{a,b,c=0}^{\infty} f_{a,b,c}\,q^a u^b s^c, \tag{30}
$$

and insist on $a \geq b \geq c$. The $b \geq c$ condition is actually already imposed by the differential equation. We demand $a \geq b$ because states at a given level $a$ can have maximum angular momentum $a$. Once this expansion is fixed, the only condition required is the normalization $f_{0,0,0} = 1$ coming from the primary state. Using this, the first few terms of the desired solution have the following small $q$ expansion:

$$
f(q,u,s) = 1 + \left(3 + \frac{\Delta_\phi(\Delta_\phi-3)}{2\Delta_{\mathcal{O}}}\right)q + \left(1 - \frac{\Delta_\phi}{2\Delta_{\mathcal{O}}}\right)qu + \left(\frac{\Delta_\phi^2}{4\Delta_{\mathcal{O}}}\right)qus + \mathcal{O}(q^2). \tag{31}
$$

As it stands, the differential equation Eq. (27) is rather complicated. We have not been able to obtain a general exact solution. In Appendix D, we consider various interesting limits of the differential equation where solutions can be easily obtained. Furthermore, encouraged by the recent AdS-integral representation of the conformal blocks in [21, 26–29], we will now look for an integral representation of the solution to Eq. (27). We will see that for zero angular potential (i.e. $u = 0$ or $y = 1$), this gives an explicit result in terms of a $_3F_2$ function (see Eq. (63)).

## 3 AdS-integral representation

In this section we derive an integral representation for the one-point thermal block. We first give a heuristic argument for the AdS-integral representation, similar to the analogous construction in $CFT_2$ of [21]. We then prove the validity of this representation, making use of the shadow formalism of [33], similar to the shadow-formalism construction of AdS geodesic Witten diagrams for flat space four-point blocks [27, 28, 43]. Finally, in Sec. 3.3, we will directly evaluate the AdS-integral representation for the case of zero angular potentials and obtain an explicit closed form result for the block.

### 3.1 Heuristic argument

Let us consider the thermal one point function of $\phi$ and its decomposition into conformal blocks

$$\left\langle \phi_{\text{cyl}}(\tau, \Omega) \right\rangle_\beta = \text{Tr}\left[ \phi_{\text{cyl}} e^{-\beta D} \right] = \sum_{\text{primary } \mathcal{O}} C_{\mathcal{O}\phi\mathcal{O}} \mathcal{F}_{\Delta_\phi; \Delta_\mathcal{O}, \ell_\mathcal{O}}(q), \tag{32}$$

where $\mathcal{F}_{\Delta_\phi; \Delta_\mathcal{O}, \ell_\mathcal{O}}(q) = q^{\Delta_\mathcal{O}}(1 + \dots)$ at small $q = e^{-\beta}$. We start by considering the contribution from a single trace operator $\mathcal{O}$. This operator is dual to a bulk field in AdS, with mass $m^2 = \Delta_\mathcal{O}(\Delta_\mathcal{O} - d)$. In first quantization, the bulk field's propagator can be computed using a particle world-line path integral.[8] The factor $q^{\Delta_\mathcal{O}} = e^{-\beta\Delta_\mathcal{O}}$ is the Boltzmann factor for this particle sitting at the origin and wrapping around the thermal cycle exactly *once*.

Now, let us present a heuristic argument/motivation for the bulk representation of the thermal blocks. Assuming a bulk cubic coupling between the bulk fields dual to $\phi$ and $\mathcal{O}$, and denoting the boundary coordinates as $x_\infty$, the full thermal one-point function can be computed by the Witten diagram

$$\left\langle \phi_{\text{cyl}}(x_\infty) \right\rangle_\beta \propto \int_{\text{thermal AdS}_{d+1}} d^{d+1}x \; \sqrt{g} G_{b\partial}^{\Delta_\phi}(x, x_\infty) G_{bb}^{\Delta_\mathcal{O}}(x, x). \tag{33}$$

Both of the propagators are thermal AdS propagators. In fact, the thermal AdS propagator $G_{bb}^{\Delta_\mathcal{O}}(x, x)$ can be obtained from that in global AdS by summing over thermal images. From a first-quantised worldline point of view, this sum over images is a sum over topologies of worldlines, organised by the number of windings around the thermal circle. The calculation of $\left\langle \phi(x_\infty) \right\rangle_\beta$ in Eq. (33) then naturally decomposes into contributions labelled by their winding around the thermal circle. This yields the sum represented pictorially in Fig. 1.[9] Since the block behaves like $q^{\Delta_\mathcal{O}}$ at small $q$, it is the *single* winding term in this sum which should be dual to the conformal block. This motivates the following proposal:

$$\mathcal{F}_{\Delta_\phi; \Delta_\phi, \ell_\mathcal{O}}(q) \sim \int_{\text{thermal AdS}_{d+1}} d^{d+1}x \; \sqrt{g} G_{b\partial}^{\Delta_\phi}(x, x_\infty) G_{bb}^{AdS, \Delta_\mathcal{O}}(x, q), \tag{34}$$

---

[8] See [44] for a detailed discussion of bulk world-line dynamics and the relation with conformal blocks.

[9] The zero winding contribution is divergent, but we omit this since it corresponds to the one-point function in global $AdS_{d+1}$, which vanishes as it is cancelled by a local counterterm.

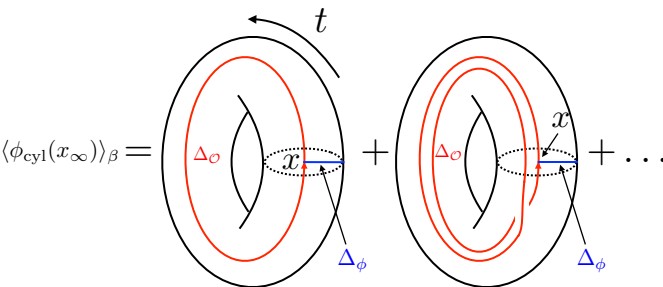

Figure 1: Bulk dual of the thermal one-point function as a sum over bulk diagrams. Here, the blue line represents the thermal AdS bulk-to-boundary propagator $G_{b\partial}^{\Delta_\phi}$. In the first diagram, the red line winding around the thermal circle once represents the $G_{bb}^{AdS,\Delta_\mathcal{O}}$ contribution, while in the second diagram, the red line winding around the thermal circle twice represents the contribution from two windings around the thermal circle. The bulk point $x$ is to be integrated over all thermal AdS.

where the propagator $G_{bb}^{AdS,\Delta_\mathcal{O}}(x,q)$ is the global AdS bulk-bulk propagator with points related by a single thermal translation. Eq. (34) is illustrated in Fig. 2. In fact, there is an alternative representation of the proposal in Eq. (34), as

$$\mathcal{F}_{\Delta_\phi;\Delta_\phi,\ell_\mathcal{O}}(q) \propto \int_{\text{AdS}_{d+1}} d^{d+1}x \sqrt{g}\; G_{b\partial}^{\text{AdS},\Delta_\phi}(x,x_\infty) G_{bb}^{\text{AdS},\Delta_\mathcal{O}}(x,q). \tag{35}$$

In this formula the integration is now over all of global AdS, while $G_{b\partial}^{\text{AdS},\Delta_\phi}(x_\infty,x)$ is the bulk-boundary propagator on global AdS. The equivalence between Eq. (34) and Eq. (35) can be shown by rewriting the thermal bulk-boundary propagator integrated over thermal AdS as (rewriting the propagator as sum over thermal images) the global bulk-boundary propagator integrated over global AdS.

To study the convergence of Eq. (35), let us consider the behavior of each object in the integrand near the boundary. We will use the global AdS metric (55) and radial coordinate $r$ so that, as $r \to \infty$, we have

$$\begin{aligned}
\sqrt{g} &\sim r^d, \\
G_{b\partial}^{\text{AdS},\Delta_\phi}(x,x_\infty) &\sim r^{\Delta_\phi-d}\left[\# + \mathcal{O}(r^{-1})\right] + r^{-\Delta_\phi}\left[\# + \mathcal{O}(r^{-1})\right], \\
G_{bb}^{\text{AdS},\Delta_\mathcal{O}}(x,q) &\sim r^{-2\Delta_\mathcal{O}},
\end{aligned} \tag{36}$$

where $\#$ are $r$-independent terms. Thus for the integral to be finite we need $\Delta_\phi < 2\Delta_\mathcal{O}$ and $d < \Delta_\phi + 2\Delta_\mathcal{O}$. When the conformal dimensions are large ($\Delta_\mathcal{O}, \Delta_\phi \gg 1$), the only nontrivial condition is $2\Delta_\mathcal{O} > \Delta_\phi$. This is easy to understand: in a saddle-point approximation the Witten diagram integral is dominated by a geodesic network with minimum total action. But there is a bulk saddle point only when $2\Delta_\mathcal{O} > \Delta_\phi$; otherwise, the bulk-to-boundary world line will drag the three point vertex all the way to the boundary.

We will provide two independent proofs of Eq. (35). In the next section we will give a proof using the shadow formalism. The will unambiguously fix the overall factor in our integral

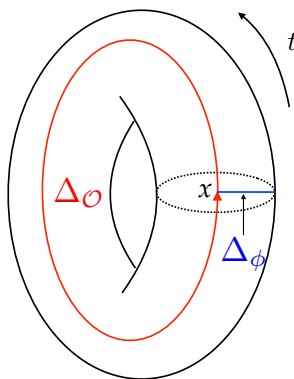

Figure 2: Bulk dual of the thermal one-point block $\mathcal{F}$. The blue line represents the bulk-to-boundary propagator for $\Delta_\phi$, and the red line represents the propagator for the $\mathcal{O}$ particle to propagate once around the thermal circle.

formula, which is so far unfixed. In Appendix E we give another proof, by showing that the RHS of Eq. (35) obeys our Casimir differential equation and has the correct low temperature behavior.

## 3.2 Construction by shadow formalism

This section includes a proof of the AdS-integral representation of the thermal one-point block. It follows essentially the same arguments laid down in [27, 28, 43] where they clarified the relation between the shadow formalism (and its projection) and the geodesic Witten diagram representation of flat space conformal blocks. In principle, this is a systematic and constructive method to build AdS-representations of any conformal block.

First, the shadow-transform of $\mathcal{O}$, denoted $\tilde{\mathcal{O}}$ (termed the "shadow operator" in [33]), is given by:

$$\tilde{\mathcal{O}}(x) \equiv \int d^d y \, \frac{1}{|x-y|^{2(d-\Delta_\mathcal{O})}} \mathcal{O}(y). \tag{37}$$

The utility of this object is that it allows us to project onto the representation generated by primary $\mathcal{O}$, via:

$$P_\mathcal{O} = \frac{1}{\mathcal{N}_\mathcal{O}} \int d^d x \, \mathcal{O}(x)|0\rangle\langle 0|\tilde{\mathcal{O}}(x), \quad \mathcal{N}_\mathcal{O} \equiv \pi^d \frac{\Gamma\left(\Delta - \frac{d}{2}\right)\Gamma\left(\frac{d}{2} - \Delta\right)}{\Gamma(\Delta)\Gamma(d-\Delta)}. \tag{38}$$

We note that with this definition, all $n$-point functions of $\tilde{\mathcal{O}}$ are related to those of $\mathcal{O}$. For example,

$$
\begin{aligned}
\langle \tilde{\mathcal{O}}(x_1)\phi(x_2)\mathcal{O}(x_3)\rangle_{\mathbb{R}^d} = {} & \int d^d y \, \frac{1}{|x_1-y|^{2(d-\Delta_\mathcal{O})}} \langle \mathcal{O}(y)\phi(x_2)\mathcal{O}(x_3)\rangle_{\mathbb{R}^d} \\
= {} & C_{\mathcal{O}\phi\mathcal{O}}|x_2-x_3|^{-\Delta_\phi} \int d^d y \, \frac{1}{|x_1-y|^{2(d-\Delta_\mathcal{O})}|y-x_2|^{\Delta_\phi}|y-x_3|^{2\Delta_\mathcal{O}-\Delta_\phi}}.
\end{aligned}
\tag{39}
$$

Performing the integral (using Eq. (2.20) of [33]) gives

$$r_C \equiv \frac{C_{\tilde{\mathcal{O}}\phi\mathcal{O}}}{C_{\mathcal{O}\phi\mathcal{O}}} = \frac{\pi^{d/2}\Gamma\left(\Delta_\mathcal{O} - \frac{d}{2}\right)\Gamma\left(\frac{d-\Delta_\phi}{2}\right)\Gamma\left(\frac{d-2\Delta_\mathcal{O}+\Delta_\phi}{2}\right)}{\Gamma\left(\frac{\Delta_\phi}{2}\right)\Gamma(d-\Delta_\mathcal{O})\Gamma\left(\Delta_\mathcal{O} - \frac{\Delta_\phi}{2}\right)}. \tag{40}$$

The thermal block is just the projection:

$$\mathcal{F}_{\Delta_\phi;\Delta_\mathcal{O}}(q) \equiv C_{\mathcal{O}\phi\mathcal{O}}^{-1} \langle \phi(x_1) P_\mathcal{O} \rangle_\beta \;=\; C_{\mathcal{O}\phi\mathcal{O}}^{-1} \sum_{i,j} (B^{-1})^{ij} \langle i | P_\mathcal{O} q^D \phi(x_1) | j \rangle_{\mathbb{R}^d}, \qquad (41)$$

where the sum is over all states in the representation generated by $\mathcal{O}$. Using the shadow representation of the projector in Eq. (38), we have:

$$\mathcal{F}_{\Delta_\phi;\Delta_\mathcal{O}}(q) \;=\; \frac{1}{C_{\mathcal{O}\phi\mathcal{O}}\mathcal{N}_\mathcal{O}} \sum_{i,j} (B^{-1})^{ij} \langle i | \left[ \int d^d x\, \mathcal{O}(x)|0\rangle\langle 0|\tilde{\mathcal{O}}(x) \right] q^D \phi(x_1)|j\rangle_{\mathbb{R}^d}$$

$$= \; \left(\mathcal{N}_\mathcal{O} C_{\mathcal{O}\phi\mathcal{O}}\right)^{-1} \int d^d x \, \langle \tilde{O}(x_q) \phi(x_1) O(x) \rangle_{\mathbb{R}^d}, \qquad (42)$$

where we have used the identity[10]

$$\sum_{i,j\in\mathcal{O},P\mathcal{O},\dots} (B^{-1})^{ij} \langle i|O(x)|0\rangle \langle\dots|j\rangle = \langle\dots O(x)|0\rangle. \qquad (45)$$

The notation $x_q$ means that $x$ is 'thermal'-translated by $q$.

All the discussions so far have been focusing on field theoretical objects in a CFT. To construct an AdS representation, we rewrite the three-point function in the integrand using an AdS-integral representation. This is given by an integral over product of three bulk-boundary propagators (assuming a $\phi\mathcal{O}^2$ coupling) [45]:

$$\int_{\text{AdS}_{d+1}} d^{d+1}y \sqrt{g}\, G_{b\partial}^{\text{AdS},d-\Delta_\mathcal{O}}(y;x') G_{b\partial}^{\text{AdS},\Delta_\phi}(y;x_1) G_{b\partial}^{\text{AdS},\Delta_\mathcal{O}}(y;x)$$

$$= c_0 C_{\tilde{O}\phi O}^{-1} \langle \tilde{O}(x')\phi(x_1)O(x)\rangle_{\mathbb{R}^d}, \qquad (46)$$

where

$$c_0 \equiv \mathcal{C}_{\Delta_\mathcal{O}} \mathcal{C}_{\Delta_\phi} \frac{\Gamma\left(\frac{\Delta_\phi}{2}\right)\Gamma\left(\frac{d-\Delta_\phi}{2}\right)\Gamma\left(\frac{d-2\Delta_\mathcal{O}+\Delta_\phi}{2}\right)\Gamma\left(\frac{-d+2\Delta_\mathcal{O}+\Delta_\phi}{2}\right)}{4\Gamma(\Delta_\mathcal{O})\Gamma\left(\frac{d}{2}-\Delta_\mathcal{O}+1\right)\Gamma\left(\Delta_\phi\right)}. \qquad (47)$$

The bulk-boundary propagator will be given explicitly in Eq. (60), but can also be written in embedding space coordinates (following the convention of [46]) as

$$G_{b\partial}^{\text{AdS},\Delta}(X;P) = \mathcal{C}_\Delta(-2P\cdot X)^{-\Delta}, \quad \mathcal{C}_\Delta \equiv \frac{\Gamma(\Delta)}{2\pi^{d/2}\Gamma(\Delta+1-d/2)}. \qquad (48)$$

---

[10] One way to show this is to first parametrize a level $n$ descendant of $|\mathcal{O}\rangle$ by $|n,\vec{n}\rangle \equiv (P\cdot\vec{n})^n|\mathcal{O}\rangle$ and denote the norm matrix at level $n$ by $B_{\vec{n}_1,\vec{n}_2}$. The LHS is then given by

$$\sum_{n=1}^\infty \sum_{\vec{n}_1,\vec{n}_2} (B^{-1})^{\vec{n}_1,\vec{n}_2} \langle n,\vec{n}_1|O(x)|0\rangle\langle\dots|n,\vec{n}_2\rangle. \qquad (43)$$

We now rewrite $O(x)|0\rangle = e^{iP\cdot x}|O\rangle = \sum_n ((iP\cdot x)^n/n!)|O\rangle = \sum_n (i^n/n!)|n,\vec{x}\rangle$. The sum now turns into

$$\sum_n \frac{i^n}{n!} \sum_{\vec{n}_1,\vec{n}_2} (B^{-1})^{\vec{n}_1,\vec{n}_2} \langle n,\vec{n}_1|n,\vec{x}\rangle\langle\dots|n,\vec{n}_2\rangle = \sum_n \frac{i^n}{n!} \sum_{\vec{n}_1,\vec{n}_2} (B^{-1})^{\vec{n}_1,\vec{n}_2} B_{\vec{n}_1,\vec{x}}\langle\dots|n,\vec{n}_2\rangle$$

$$= \sum_n \frac{i^n}{n!} \langle\dots|n,\vec{x}\rangle = \sum_n \frac{i^n}{n!} \langle\dots|(P\cdot\vec{x})^n|\mathcal{O}\rangle = \langle\dots\mathcal{O}(x)|0\rangle. \qquad (44)$$

We can use the symmetry of the propagator to deduce that

$$G_{b\partial}^{\mathrm{AdS},d-\Delta_{\mathcal{O}}}(y;x_q) = G_{b\partial}^{\mathrm{AdS},d-\Delta_{\mathcal{O}}}(y_{q^{-1}};x), \tag{49}$$

and thus

$$\langle \tilde{O}(x_q)\phi(x_1)O(x)\rangle_{\mathbb{R}^d}$$
$$= c_0^{-1} C_{\tilde{O}\phi\mathcal{O}} \int_{\mathrm{AdS}_{d+1}} d^{d+1}y \sqrt{g}\, G_{b\partial}^{\mathrm{AdS},d-\Delta_{\mathcal{O}}}(y_q;x)G_{b\partial}^{\mathrm{AdS},\Delta_\phi}(y;x_1)G_{b\partial}^{\mathrm{AdS},\Delta_{\mathcal{O}}}(y;x). \tag{50}$$

Substituting this AdS-representation of the three-point function back into the integrand, the conformal block becomes

$$\mathcal{F}_{\Delta_\phi;\Delta_{\mathcal{O}}}(q)$$
$$= r_C\,(\mathcal{N}_{\mathcal{O}}c_0)^{-1} \int_{\mathrm{AdS}_{d+1}} d^{d+1}y \sqrt{g}\, G_{b\partial}^{\mathrm{AdS},\Delta_\phi}(y;x_1)\left[\int d^d x\, G_{b\partial}^{\mathrm{AdS},d-\Delta_{\mathcal{O}}}(y_q;x)G_{b\partial}^{\mathrm{AdS},\Delta_{\mathcal{O}}}(y;x)\right]$$
$$= r_C\,(\mathcal{N}_{\mathcal{O}}c_0(d-2\Delta_{\mathcal{O}}))^{-1} \int_{\mathrm{AdS}_{d+1}} d^{d+1}y \sqrt{g}\, G_{b\partial}^{\mathrm{AdS},\Delta_\phi}(y;x_1)\Omega_{\Delta_{\mathcal{O}}}(y,y_q), \tag{51}$$

where we have used the split representation of the AdS harmonic function

$$\Omega_\Delta(y_1,y_2) \equiv G_{bb}^{\mathrm{AdS},\Delta}(y_1,y_2) - G_{bb}^{\mathrm{AdS},d-\Delta}(y_1,y_2)$$
$$= (d-2\Delta) \int d^d x\, G_{b\partial}^{\mathrm{AdS},\Delta}(y_1;x)G_{b\partial}^{\mathrm{AdS},d-\Delta}(y_2;x). \tag{52}$$

The AdS harmonic function is defined as the regular (at coincident point) Green's function. It is a particular linear combination of the Green's function of a bulk field of dimension $\Delta$ and a bulk field of dimension $d-\Delta$. The shadow formalism instructs us to further project, either using monodromy projection or utilizing some clever integration contour, onto just the block associated with operator with dimension $\Delta$. This projection in terms of AdS representation amounts to replacing the bulk harmonic function $\Omega$ with the bulk-bulk propagator $G_{bb}$ used for a field dual to an operator of dimension $\Delta$, and so finally we obtain

$$\mathcal{F}_{\Delta_\phi;\Delta_{\mathcal{O}}}(q) = r_C\,(\mathcal{N}_{\mathcal{O}}c_0(d-2\Delta_{\mathcal{O}}))^{-1} \int_{\mathrm{AdS}_{d+1}} d^{d+1}y \sqrt{g}\, G_{b\partial}^{\mathrm{AdS},\Delta_\phi}(y;x_\infty)G_{bb}^{\mathrm{AdS},\Delta_{\mathcal{O}}}(y,q). \tag{53}$$

This is exactly the proposal for the thermal one-point block in Eq. (35). The boundary condition at small $q$ is just the statement that as $\beta \to \infty$ we expect the bulk-bulk propagator to vanish as $e^{-\Delta_{\mathcal{O}}\beta} = q^{\Delta_{\mathcal{O}}}$. If we had used the harmonic function, we would have obtained in addition a $e^{-(d-\Delta_{\mathcal{O}})\beta} = q^{d-\Delta_{\mathcal{O}}}$ behavior at small $q$.

There are a few observations and comments to be made:

1. This construction unambiguously fixes the overall coefficient of the bulk AdS-representation. More explicitly,

$$\int d^{d+1}y\, G_{b\partial}^{\Delta_\phi}(y;x_\infty)G_{bb}^{\Delta_{\mathcal{O}}}(y,q)$$
$$= \mathcal{C}_{\Delta_{\mathcal{O}}}\mathcal{C}_{\Delta_\phi}\left[\frac{\pi^{d/2}\Gamma\left(\frac{\Delta_\phi}{2}\right)^2\Gamma\left(\Delta_{\mathcal{O}}-\frac{\Delta_\phi}{2}\right)\Gamma\left(-\frac{d}{2}+\Delta_{\mathcal{O}}+\frac{\Delta_\phi}{2}\right)}{2\Gamma(\Delta_{\mathcal{O}})^2\Gamma(\Delta_\phi)}\right]\mathcal{F}_{\Delta_\phi;\Delta_{\mathcal{O}}}(q). \tag{54}$$

This overall factor will be verified explicitly in the next section, when we compute the AdS integral.

2. In fact, without resolving to an AdS representation of the three-point function, one should be able to directly compute the conformal block using the last line of Eq. (42) by performing the integral over the three-point function. To obtain the physical conformal blocks, we need to further perform the monodromy projection [33]. It would be interesting to do this calculation directly and obtain the explicit conformal blocks.

3. Finally, we used the regular Witten diagram representation of the CFT three-point function in our construction. We could have instead used a "geodesic Witten diagram" bulk representation, by writing a boundary three-point function as a bulk geodesic integral. However, since we have only inserted one boundary external operator $\phi$, and the rest of the calculation involves integrating over the insertion point of $\mathcal{O}$ (or $\tilde{\mathcal{O}}$), the final form of the integral will not involve an integral over fixed geodesics (as in the flat space four-point block case) but rather over geodesics anchored on at least one boundary point integrated over the boundary. This might prove useful for some purposes, but at the moment it seems like a more cumbersome representation than the one given above.

4. Just as the shadow formalism was useful in constructing the AdS-integral representation of the flat space spinning conformal blocks [27,28,43], it will be interesting in the future to generalize the discussions in this section to operators with spin.

### 3.3 Explicit AdS integral representation

We shall evaluate the RHS of Eq. (35) in the case of zero angular potential. We use global coordinates $(t, r, \Omega_{d-1})$ on global $\text{EAdS}_{d+1}$ with metric

$$ds^2 = (1 + r^2)dt^2 + (1 + r^2)^{-1}dr^2 + r^2 d\Omega_{d-1}^2. \tag{55}$$

We must integrate the product of a bulk-to-bulk propagator and a bulk-to-boundary propagator. The AdS bulk-to-bulk propagator is (using embedding coordinates $X$ and $Y$):[11]

$$G_{bb}^{\text{AdS},\Delta}(X,Y) = \mathcal{C}_\Delta u^{-\Delta} {}_2F_1\left(\Delta, \frac{2\Delta - d + 1}{2}; 2\Delta - d + 1; -4u^{-1}\right), \quad u = (X - Y)^2. \tag{56}$$

We need to compute the geodesic distance between a point $X$ and its thermal translation $X_\beta$. Using the relations between the embedding coordinates $X$ and the global coordinates

$$
\begin{aligned}
U &= \sqrt{1 + r^2}\sinh t, \quad V = \sqrt{1 + r^2}\cosh t, \\
X_i &= r\Omega_i, \quad i = 1, \dots, d, \quad \sum_{i=1}^{d} \Omega_i^2 = 1,
\end{aligned}
\tag{57}
$$

where $\sum_{i=1}^{d} X_i^2 + U^2 - V^2 = -1$ we have

$$X \cdot X_\beta = -(1 + r^2)\cosh(\beta) + r^2, \tag{58}$$

so we obtain

$$u = \frac{(1-q)^2}{q}(1 + r^2), \quad -\frac{4}{u} = -\frac{4q}{(1-q)^2(1 + r^2)}. \tag{59}$$

We see that $G_1^\Delta(x, q) \equiv G_{bb}^{\text{AdS},\Delta}(X, X_\beta)$ is a function only of $r$, not $t$ and $\Omega_i$. Thus the bulk integral over $t$ and $\Omega$ can be performed by just focusing on the bulk-to-boundary part. To proceed, recall that the bulk-to-boundary propagator is given by

$$G_{b\partial}^{\text{AdS},\Delta_\phi}(t, r, \Omega; t_\infty, \Omega_\infty) = \mathcal{C}_{\Delta_\phi} 2^{-\Delta_\phi}\left[\sqrt{1 + r^2}\cosh(t - t_\infty) - r\cos\Theta(\Omega, \Omega_\infty)\right]^{-\Delta_\phi}. \tag{60}$$

---

[11]We will follow the conventions in Appendix B.1 of [46].

$\Theta$ is the relative angle between $\hat{\Omega}_\infty$ and $\hat{\Omega}$ on the sphere $S_{d-1}$. Collecting all the ingredients, the integral in Eq. (35) is explicitly given as:

$$
\begin{aligned}
I &= \mathcal{C}_{\Delta_\mathcal{O}} \mathcal{C}_{\Delta_\phi} 2^{-\Delta_\phi} \int_{-\infty}^\infty dt\, dr\, d^{d-1}\Omega\; r^{d-1} \left[ \frac{(1-q)^2}{q}(1+r^2) \right]^{-\Delta_\mathcal{O}} \\
&\quad \times {}_2F_1\left[ \Delta_\mathcal{O}, \frac{2\Delta_\mathcal{O}-d+1}{2}; 2\Delta_\mathcal{O}-d+1; -\frac{4q}{(1-q)^2(1+r^2)} \right] \\
&\quad \times \left[ \sqrt{1+r^2}\cosh(t-t_\infty) - r\cos\Theta(\Omega,\Omega_\infty) \right]^{-\Delta_\phi}.
\end{aligned}
\tag{61}
$$

For the integral over $\Omega$, by spherical symmetry, we can rotate the axis parallel to $\Omega_\infty$, and the integral reduces to one only over an angular variable $\Theta$ from 0 to $\pi$. Including the appropriate angular measure, this part of the integral becomes[12]

$$
\begin{aligned}
J &\equiv \int_{-\infty}^\infty dt\, d^{d-1}\Omega \left[ \sqrt{1+r^2}\cosh(t-t_\infty) - r\cos\Theta(\Omega,\Omega_\infty) \right]^{-\Delta_\phi} \\
&= \frac{2\pi^{\frac{d-1}{2}}}{\Gamma\left(\frac{d-1}{2}\right)} \int_{-\infty}^\infty dt \int_0^\pi d\Theta\; \sin^{d-2}\Theta \left[ \sqrt{1+r^2}\cosh t - r\cos\Theta \right]^{-\Delta_\phi} \\
&= \frac{2\pi^{\frac{d+1}{2}}}{\Gamma\left(\frac{d}{2}\right)} \frac{\Gamma\left(\frac{\Delta_\phi}{2}\right)}{\Gamma\left(\frac{1}{2}(\Delta_\phi+1)\right)} \left[1+r^2\right]^{-\frac{1}{2}\Delta_\phi} {}_2F_1\left( \frac{\Delta_\phi}{2}, \frac{\Delta_\phi}{2}; \frac{d}{2}; \frac{r^2}{r^2+1} \right).
\end{aligned}
\tag{62}
$$

With this part of the integral done, the full integral is then reduced to

$$
\begin{aligned}
I &= \mathcal{C}_{\Delta_\mathcal{O}} \mathcal{C}_{\Delta_\phi} 2^{-\Delta_\phi} \frac{2\pi^{\frac{d+1}{2}}}{\Gamma\left(\frac{d}{2}\right)} \frac{\Gamma\left(\frac{\Delta_\phi}{2}\right)}{\Gamma\left(\frac{1}{2}(\Delta_\phi+1)\right)} q^{\Delta_\mathcal{O}}(1-q)^{-2\Delta_\mathcal{O}} \times \int_0^\infty dr\, r^{d-1}\left[1+r^2\right]^{-\frac{1}{2}\Delta_\phi-\Delta_\mathcal{O}} \\
&\quad \times {}_2F_1\left( \frac{\Delta_\phi}{2}, \frac{\Delta_\phi}{2}; \frac{d}{2}; \frac{r^2}{r^2+1} \right) {}_2F_1\left[ \Delta_\mathcal{O}, \frac{2\Delta_\mathcal{O}-d+1}{2}; 2\Delta_\mathcal{O}-d+1; -\frac{4q}{(1-q)^2(1+r^2)} \right] \\
&= \mathcal{C}_{\Delta_\mathcal{O}} \mathcal{C}_{\Delta_\phi} \left[ \frac{\pi^{\frac{d}{2}}\Gamma\left(\Delta_\mathcal{O}-\frac{\Delta_\phi}{2}\right)\Gamma\left(\frac{1}{2}(\Delta_\phi-d)+\Delta_\mathcal{O}\right)\Gamma\left(\frac{\Delta_\phi}{2}\right)^2}{2\Gamma(\Delta_\mathcal{O})^2\Gamma(\Delta_\phi)} \right] q^{\Delta_\mathcal{O}}(1-q)^{-2\Delta_\mathcal{O}} \\
&\quad \times {}_3F_2\left( -\frac{d}{2}+\Delta_\mathcal{O}+\frac{1}{2}, \Delta_\mathcal{O}-\frac{\Delta_\phi}{2}, -\frac{d}{2}+\frac{\Delta_\phi}{2}+\Delta_\mathcal{O}; \Delta_\mathcal{O}, -d+2\Delta_\mathcal{O}+1; -\frac{4q}{(q-1)^2} \right).
\end{aligned}
\tag{63}
$$

This is an explicit representation of the thermal one-point block. Note that the overall factor (i.e. square bracket terms in the second to last line) agrees with that coming from the shadow-block construction (see Eq. (54)). Furthermore, we have verified that the last line agrees with the explicit evaluation of the blocks up to order $q^4$ given in Eq. (119) for $d=3$. Note that in performing various integrals we have assumed that $\Delta_\phi + 2\Delta_\mathcal{O} > d$ and $\Delta_\mathcal{O} > \frac{\Delta_\phi}{2}$. Together, these conditions can be combined to give $\Delta_\mathcal{O} > \frac{d}{4}$.

There are a few simple limits of the blocks that are of interest:

- The simplest one is the high temperature limit: When we take the limit $q \to 1$, the blocks behave as

$$
(1-q)^{-\max(\Delta_\phi, d-\Delta_\phi, d-1)}.
\tag{64}
$$

---

[12] See Appendix F for more details on carrying out the integrals in Eq. (62)-(63).

- Another simple limit is the $\Delta_\phi \to 0$ limit, where it should reproduce the character. Indeed, setting $\Delta_\phi = 0$ in (63) gives

$$
\mathcal{F} = q^{\Delta_\mathcal{O}}(1-q)^{-2\Delta_\mathcal{O}} \, {}_3F_2\left(-\frac{d}{2}+\Delta_\mathcal{O}+\frac{1}{2}, \Delta_\mathcal{O}, -\frac{d}{2}+\Delta_\mathcal{O}; \Delta_\mathcal{O}, -d+2\Delta_\mathcal{O}+1; -\frac{4q}{(q-1)^2}\right)
$$
$$
= q^{\Delta_\mathcal{O}}(1-q)^{-2\Delta_\mathcal{O}} \, {}_2F_1\left(-\frac{d}{2}+\Delta_\mathcal{O}+\frac{1}{2}, -\frac{d}{2}+\Delta_\mathcal{O}; -d+2\Delta_\mathcal{O}+1; -\frac{4q}{(q-1)^2}\right). \tag{65}
$$

At this point we can use the following identity for the hypergeometric function when $|z| < 1$:

$$
{}_2F_1\left[a, b; 2b; \frac{4z}{(z+1)^2}\right] = (z+1)^{2a} \, {}_2F_1\left[a, a-b+\frac{1}{2}; b+\frac{1}{2}; z^2\right] \tag{66}
$$

to rewrite the expression for the block as

$$
q^{\Delta_\mathcal{O}}(1-q)^{-d} \, {}_2F_1\left(-\frac{d}{2}+\Delta_\mathcal{O}, 0; -\frac{d}{2}+\Delta_\mathcal{O}+1; q^2\right) = q^{\Delta_\mathcal{O}}(1-q)^{-d}. \tag{67}
$$

This is precisely the character of the conformal algebra, as expected.

- Finally, we note that in $d = 2$ case our formula reduces to the square of $_2F_1$, which matches precisely previously obtained expressions for thermal blocks in two dimensional CFTs (given in e.g. [18]).

To understand more complicated limits, we first note that our block $F$ satisfies the differential equation

$$
\begin{aligned}
0 = & \; \mathcal{F}^{(3)}(q) + \frac{(d+4)q + d - 2}{(q-1)q}\mathcal{F}''(q) \\
& + q^{-2}\left[-m_\mathcal{O}^2 + \frac{q}{(q-1)^2}\left((d+1)(d-2+2q) - m_\phi^2\right)\right]\mathcal{F}'(q) \\
& - \frac{1}{2}(q-1)^{-3}q^{-3}(q+1)\left[2(q-1)^2 m_\mathcal{O}^2 + (d-1)m_\phi^2 q\right]\mathcal{F}(q), \tag{68}
\end{aligned}
$$

where $m_\phi^2 \equiv \Delta_\phi(\Delta_\phi - d)$ and $m_\mathcal{O}^2 \equiv \Delta_\mathcal{O}(\Delta_\mathcal{O} - d)$. With this equation at hand, let us study two other limits.

### 3.3.1 WKB limit

We first consider a particular WKB limit, where the block will be given by the action of a heavy particle following a geodesic in AdS. We will let $\rho \equiv \Delta_\phi/\Delta_\mathcal{O}$ and take $\Delta_\mathcal{O} \to \infty$ with fixed $\rho$. Inserting the ansatz $\mathcal{F} = q^{\Delta_\mathcal{O}} e^{-\Delta_\phi G}$ into the differential equation we obtain, at large $\Delta_\mathcal{O}$,

$$
0 = \left[G'(q) - \frac{1}{q\rho}\right]\left[qG'(q)^2 - 2\rho^{-1}G'(q) - \frac{1}{(q-1)^2}\right]. \tag{69}
$$

Two of the solutions for $G$ behave as $\log q$ near $q \to 0$, so are discarded. The log-free solution is

$$
G' = \frac{1}{q\rho} + \frac{\sqrt{q^2 + (\rho^2 - 2)q + 1}}{(q-1)q\rho}, \tag{70}
$$

and gives the asymptotic behaviour of the conformal block in this limit. Note that the $d$-dependence drops out, so this reduces to the same equation as in $d = 2$ discussed in [21]. Indeed, this result reproduces the AdS-bulk geodesic computation since when there are no angular potentials the $\text{AdS}_{d+1}$ geodesic computation is also independent of $d$.[13] This WKB limit can also be studied using the Casimir equation, as described in Appendix D.1.

---

[13]We thank Henry Maxfield for making the observation, and bringing to our attention the fact that the $\text{AdS}_{d+1}$ geodesic computation is independent of $d$.

### 3.3.2 Large $\Delta_{\mathcal{O}}$ limit

Here we study the limit of large $\Delta_{\mathcal{O}}$ with $\Delta_{\phi}$ fixed. This will be useful for deriving asymptotic OPE coefficients in Sec. 4.3. Let us start with the ansatz $\mathcal{F}(q) = q^{\Delta_{\mathcal{O}}} f(q)$. Taking the limit $\Delta_{\mathcal{O}} \to \infty$, we obtain the leading differential equation:

$$0 = d f(q) + (q-1) f'(q) \quad \Rightarrow F(q) \to q^{\Delta_{\mathcal{O}}}(1-q)^{-d}, \tag{71}$$

which is just the character. To study the first correction, we substitute the ansatz $F(q) = q^{\Delta_{\mathcal{O}}} \left( f + \Delta_{\mathcal{O}}^{-1} g + \dots \right)$ to obtain the differential equation for $g$

$$0 = g'(q) + \frac{d}{q-1} g - \frac{1}{2}(1-q)^{-d-2} m_{\phi}^2. \tag{72}$$

With boundary condition $g \sim q^1$ near $q = 0$, we have

$$g = \frac{m_{\phi}^2}{2} \frac{q}{(1-q)^{d+1}}. \tag{73}$$

For $d = 3$, this can also be derived using the Casimir differential equation, as is done in Appendix D.2.

Finally, note that this approximation is valid when $f(q) \gg g(q)/\Delta_{\mathcal{O}}$, which occurs when $\beta \Delta_{\mathcal{O}} \gg 1$.

## 4 An application: asymptotics of OPE coefficients

In this section we will study the asymptotic behaviour of the light-heavy-heavy OPE coefficients for primary operators. We will start by reviewing a simple and well-known estimate for the high energy density of states in a CFT. We will then study the high temperature limit of thermal one-point functions in CFT$_d$ for $d > 2$. This leads to an asymptotic expression for the average value of a light-heavy-heavy OPE coefficient, averaged over the dimension of the heavy operator. We will first consider the average over *all* operators, before using our knowledge of conformal blocks to compute the average over primary operators. Our final result for the average over primary operators will be the same as for the average over all operators. Note that in this section we shall use

$$\langle \phi \rangle_{\beta} \equiv \frac{1}{Z(\beta)} \text{Tr}\left[ \phi \; e^{-\beta D} \right] \tag{74}$$

to denote the *normalized* cylinder thermal one-point function.

### 4.1 Density of states in general dimensions

Before studying one-point functions, we first need to understand the asymptotic density of states of a CFT in $d$ dimensions.[14] We will write the finite temperature partition function on $S^{d-1}$ as

$$Z(\beta) = \sum_{\mathcal{O}} e^{-\frac{\beta}{R} \Delta_{\mathcal{O}}} = \int d\Delta \, \rho(\Delta) e^{-\frac{\beta}{R} \Delta}, \tag{75}$$

where $\rho(\Delta) \equiv \sum_{\mathcal{O}} \delta(\Delta - \Delta_{\mathcal{O}})$ is the density of states. Here $R$ is the radius of the sphere $S^{d-1}$. We will not set $R = 1$ in this section, in order to emphasize the scaling behaviour of our results.

---

[14]The analysis in this section is not new; see, e.g. section 4.3 of [47] for a nice summary.

In the thermodynamic limit, the free energy must be proportional to the spatial volume $A_{d-1}R^{d-1}$, where $A_{d-1} = 2\pi^{d/2}/\Gamma\left(\frac{d}{2}\right)$ is the area of the unit $(d-1)$-sphere. Scale invariance then fixes the free energy at high temperature to be

$$F = -\tilde{c}A_{d-1}R^{d-1}\beta^{-d} + \dots, \tag{76}$$

where $\tilde{c}$ is a theory-dependent dimensionless constant. In $d = 2$, $\tilde{c}$ is the central charge. In higher dimensions, $\tilde{c}$ is best understood as a "normalized entropy density" [48], which is generally not equal to a coefficient appearing in stress tensor two or three point functions. We can then perform an inverse Laplace transform of the partition function to obtain the density of states

$$\rho(\Delta) \approx \frac{1}{2\pi i R} \oint d\beta \, e^{\tilde{c}A_{d-1}\left(\frac{\beta}{R}\right)^{1-d} + \frac{\beta}{R}\Delta}. \tag{77}$$

For large $\Delta$, this integral can be evaluated in a saddle point approximation. The saddle is at $\frac{\beta}{R} = \left(\frac{(d-1)\tilde{c}A_{d-1}}{\Delta}\right)^{\frac{1}{d}}$ and the result is

$$\rho(\Delta) \approx \frac{1}{\sqrt{2\pi d(d-1)\tilde{c}A_{d-1}}}\left(\frac{(d-1)\tilde{c}A_{d-1}}{\Delta}\right)^{\frac{d+1}{2d}} e^{d(d-1)^{\frac{1}{d}-1}(\tilde{c}A_{d-1})^{\frac{1}{d}}\Delta^{\frac{d-1}{d}}}. \tag{78}$$

### 4.2 Asymptotics for generic operators

The (normalized) thermal expectation value of an operator is defined as

$$\langle\phi\rangle_\beta = \frac{1}{Z(\beta)}\sum_{\mathcal{O}}\langle\mathcal{O}|\phi|\mathcal{O}\rangle e^{-\frac{\beta}{R}\Delta_{\mathcal{O}}}. \tag{79}$$

Even though $\phi$ is a scalar, the sum over $\mathcal{O}$ includes all states in the theory, including those with large spin. However, the existence of a thermodynamic limit implies that the sum in Eq. (79) is dominated by states with small spin, so we can neglect the large spin states in this expression. The reason is the following. Eq. (79) can be viewed as an expectation value in an ensemble with fixed temperature and zero angular potential. We can also consider another ensemble with fixed temperature and zero angular momentum, rather than zero angular potential. In the high temperature limit, the equivalence of canonical and microcanonical ensembles implies that the system with zero angular momentum potential and zero angular momentum should be the same. More precisely, in the thermodynamic limit only those states with

$$\ell_{\mathcal{O}}/\Delta_{\mathcal{O}} \ll 1, \text{ as } \Delta_{\mathcal{O}} \gg 1 \tag{80}$$

will dominate the sum. So the sum in Eq. (79) will be dominated by operators $\mathcal{O}$ with small spin.[15]

We can now rewrite the RHS as

$$\langle\phi\rangle_\beta = \frac{1}{Z(\beta)}\int d\Delta \, T_\phi(\Delta) e^{-\frac{\beta}{R}\Delta}, \tag{81}$$

where

$$T_\phi(\Delta) \equiv \sum_{\mathcal{O}}\langle\mathcal{O}|\phi|\mathcal{O}\rangle \, \delta(\Delta - \Delta_{\mathcal{O}}) \tag{82}$$

is the "density of OPE Coefficients," in analogy with the density of states.

---

[15]We are grateful to N. Lashkari for discussions related to this point.

At high temperature, on general grounds we expect Eq. (79) to behave as

$$\langle \phi \rangle_\beta \approx \alpha_\phi \left(\frac{\beta}{R}\right)^{-\Delta_\phi}, \quad \text{as } \beta \to 0. \tag{83}$$

The constant $\alpha_\phi$ is dimensionless and depends on the operator $\phi$ as well as the theory being studied. Eq. (83) is simply the statement that in the high temperature limit the thermal one point function on $S^{d-1}$ will go over to that on $\mathbb{R}^{d-1}$, which is proportional to $\beta^{-\Delta_\phi}$ by scale invariance. In general, it is possible for the coefficient $\alpha_\phi$ to vanish, but we generically expect $\alpha_\phi \neq 0$ unless it is set to zero by some symmetry. We will therefore just assume that $\alpha_\phi \neq 0$, so that Eq. (83) determines the high energy behaviour.

Combining Eq. (83) and Eq. (79), we get

$$\alpha_\phi \left(\frac{\beta}{R}\right)^{-\Delta_\phi} e^{\tilde{c}A_{d-1}(\beta/R)^{1-d}} = \int d\Delta\, T_\phi(\Delta) e^{-\frac{\beta}{R}\Delta}, \tag{84}$$

which can be inverted as before to get

$$T_\phi(\Delta) = \frac{\alpha_\phi}{2\pi i R} \oint d\beta \left(\frac{\beta}{R}\right)^{-\Delta_\phi} e^{\tilde{c}A_{d-1}\left(\frac{\beta}{R}\right)^{1-d}+\frac{\beta}{R}\Delta}, \tag{85}$$

which is dominated again by the saddle point $\frac{\beta}{R} = \left(\frac{(d-1)\tilde{c}A_{d-1}}{\Delta}\right)^{\frac{1}{d}}$ when $\Delta$ is large. The result is

$$T_\phi(\Delta) \approx \frac{\alpha_\phi}{\sqrt{2\pi d(d-1)\tilde{c}A_{d-1}}} \left(\frac{(d-1)\tilde{c}A_{d-1}}{\Delta}\right)^{\frac{d+1}{2d}} e^{d(d-1)^{\frac{1}{d}-1}(\tilde{c}A_{d-1})^{\frac{1}{d}}\Delta^{\frac{d-1}{d}}} \left(\frac{\Delta}{(d-1)\tilde{c}A_{d-1}}\right)^{\frac{\Delta_\phi}{d}}. \tag{86}$$

Combining this with the result for the density of states, we can deduce the average value of the three-point function coefficient:

$$\overline{\langle \mathcal{O}| \phi |\mathcal{O}\rangle} \equiv \frac{T_\phi(\Delta)}{\rho(\Delta)} \approx \alpha_\phi \left(\frac{\Delta}{(d-1)\tilde{c}A_{d-1}}\right)^{\frac{\Delta_\phi}{d}}. \tag{87}$$

This is simply the thermal expectation value of $\phi$ evaluated at the saddle point. The corrections to this equation will depend on the details of the theory, and in particular on the finite size corrections appearing in Eq. (83). This formula also makes an appearance as the diagonal part of the Eigenstate Thermalization Hypothesis, as described in [34, 49, 50].

### 4.3 Asymptotics for primaries

We can now take this one step further, and write the thermal one-point function of a scalar operator as a sum only over primary operators, along with an appropriate conformal block:

$$\langle \phi \rangle_\beta = \frac{1}{Z(\beta)} \sum_{primaries\, \mathcal{O}} \langle \mathcal{O}| \phi |\mathcal{O}\rangle\, \mathcal{F}_{\Delta_\phi,\Delta_\mathcal{O}}(q) e^{-\frac{\beta}{R}\Delta_\mathcal{O}}. \tag{88}$$

Note that, as discussed above, the existence of a thermodynamic limit means that this sum will be dominated by operators $\mathcal{O}$ with small spin.[16] So we can safely use the scalar block $\mathcal{F}_{\Delta_\phi,\Delta_\mathcal{O}}$ in this expression.[17]

---

[16]In particular, we meant that the spin-energy ratio satisfies Eq. 80. Moreover, given such a primary operator, its descendants will also satisfy Eq. 80 since their spins do not scale as $\Delta_\mathcal{O}$. Since the number of descendants do not scale exponentially as their energy is taken to be large, the thermodynamics argument still applies.

[17]Note that the conformal block with non-zero (but small) $\ell_\mathcal{O}$ is the same as the scalar block in the limit $\Delta_\mathcal{O} \gg 1$. This is because the conformal block with internal spin satisfies the same differential equation as that without spin, but with a new Casimir value of $\mathcal{O}$ given by $\Delta_\mathcal{O}(\Delta_\mathcal{O}-d)+\ell_\mathcal{O}(\ell_\mathcal{O}+d-2)$. In the limit of $\ell_\mathcal{O} \ll \Delta_\mathcal{O}$, this reduces to the same Casimir as the scalar case.

We again take the large temperature limit of the LHS and write the RHS as an integral:

$$\alpha_\phi \left(\frac{\beta}{R}\right)^{-\Delta_\phi} e^{\tilde{c}A_{d-1}(\beta/R)^{1-d}} = \int d\Delta\, T_\phi^{prim}(\Delta) \mathcal{F}_{\Delta_\phi,\Delta}(q) e^{-\frac{\beta}{R}\Delta}, \tag{89}$$

where $T_\phi^{prim}(\Delta)$ is the sum of OPE coefficients $\langle\mathcal{O}|\,\phi\,|\mathcal{O}\rangle$ for all the primaries $\mathcal{O}$ of dimension $\Delta$, defined as in Eq. (82). As in the previous cases, the integral will be dominated by large $\Delta$. We studied the conformal blocks in this limit in Section 3.3.2; restoring factors of $R$, they go like $\mathcal{F}_{\Delta_\phi,\Delta}(q) \sim \left(\frac{\beta}{R}\right)^{-d}$ at high temperature. We can then invert this expression using a saddle point exactly as before to find an asymptotic formula for $T_\phi^{prim}(\Delta)$. The result is exactly as in (86) except that, because of the contribution of the blocks, we must shift the external dimension $\Delta_\phi$ by $-d$.

Finally, in order to compute the average value of the OPE coefficient we need the asymptotic density of primary states $\rho^{prim}(\Delta)$. The computation is exactly as in the computation of the density of states, except that we now have to invert

$$e^{\tilde{c}A_{d-1}\left(\frac{\beta}{R}\right)^{1-d}} = \int d\Delta\, \rho^{prim}(\Delta)\left(\frac{\beta}{R}\right)^{-d} e^{-\frac{\beta}{R}\Delta}. \tag{90}$$

Here the factor of $\left(\frac{\beta}{R}\right)^{-d}$ comes from the behavior of the conformal characters at high temperature. We can again invert this using a saddle point approximation, and use this to find an expression for the average primary operator coefficient. The result is

$$\overline{\langle\mathcal{O}|\,\phi\,|\mathcal{O}\rangle}_{primary} \equiv \frac{T_\phi^{prim}(\Delta)}{\rho^{prim}(\Delta)} \approx \alpha_\phi \left(\frac{\Delta}{(d-1)\tilde{c}A_{d-1}}\right)^{\frac{\Delta_\phi}{d}}. \tag{91}$$

Note that the extra contribution from the blocks appearing in (89) exactly cancels that from the characters in (90). The result is an asymptotic expression for the average primary operator OPE coefficient which exactly matches that for the average over all OPE coefficients given in (87).[18]

## Acknowledgements

We thank B. Chen, A. Dymarsky, D. Harlow, P. Kraus, N. Lashkari, H. Maxfield, A. Parnachev, E. Perlmutter, D. Poland and S. Shatashvili for discussions. We acknowledge the support of the Natural Sciences and Engineering Research Council of Canada (NSERC), funding reference number SAPIN/00032-2015. This work was supported in part by a grant from the Simons Foundation (385602, A.M.). G. N. is supported by Simons Foundation Grant to HMI under the program "Targeted Grants to Institutes". Jie-qiang Wu is supported by Massachusetts Institute of Technology and Simons foundation it from qubit collaboration. Y. G. is supported by the National Science and Engineering Council of Canada, the Fonds de recherche du Québec: Nature et technologies and by a Walter C. Sumner Memorial Fellowship. This work was performed in part at the Aspen Center for Physics, which is supported by National Science Foundation grant PHY-1607611.

---

[18]It is interesting to contrast this situation with what happens in $d = 2$. In that case, when one computes the average OPE coefficient of Virasoro primaries, one obtains a slightly different formula from the average over all OPE coefficients [20]: in particular, the central charge is shifted by $c \to c-1$. This reflects a qualitative difference between Virasoro blocks and global conformal blocks. For example, at high energy the number of states in the Virasoro Verma module grows like that of a CFT with $c = 1$.

# A Conformal algebra and representations: notations and conventions

Consider a Euclidean $CFT_d$ on $\mathbb{R}^d$ with coordinates $x^\mu$ ($\mu = 1, \dots d$) and work in radial quantization. The notation used is the same as in [4]. The conformal algebra is composed of translations $P_\mu$, dilatations $D$, special conformal transformations $K_\mu$, and rotations $M_{\mu\nu}$, which satisfy the following commutation relations

$$
\begin{aligned}
\left[D, P_\mu\right] &= P_\mu, \quad \left[D, K_\mu\right] = -K_\mu, \quad \left[K_\mu, P_\nu\right] = 2\delta_{\mu\nu}D - 2M_{\mu\nu}, \\
\left[M_{\mu\nu}, P_\rho\right] &= \delta_{\nu\rho}P_\mu - \delta_{\mu\rho}P_\nu, \quad \left[M_{\mu\nu}, K_\rho\right] = \delta_{\nu\rho}K_\mu - \delta_{\mu\rho}K_\nu, \\
\left[M_{\mu\nu}, M_{\rho\sigma}\right] &= \delta_{\nu\rho}M_{\mu\sigma} - \delta_{\mu\rho}M_{\nu\sigma} + \delta_{\nu\sigma}M_{\rho\mu} - \delta_{\mu\sigma}M_{\rho\nu},
\end{aligned}
\tag{92}
$$

while other commutators vanish. The commutators involving only the $M_{\mu\nu}$ are recognized to be the $SO(d)$ algebra. Note that the generators satisfy $D^\dagger = D$, $M_{\mu\nu}^\dagger = -M_{\mu\nu}$, $P_\mu^\dagger = K_\mu$, and $K_\mu^\dagger = P_\mu$.

The states in a CFT are classified either as primaries or descendants and are labelled by their dilatation eigenvalue $\Delta$ and their $SO(d)$ representation $S_{\mu\nu}$. A primary state $|\mathcal{O}\rangle$ satisfies $D|\mathcal{O}\rangle = \Delta|\mathcal{O}\rangle$, $M_{\mu\nu}|\mathcal{O}\rangle = S_{\mu\nu}|\mathcal{O}\rangle$ (spin indices suppressed) and $K_\mu|\mathcal{O}\rangle = 0$. The rest of the states are descendants, which are build out of primaries in highest-weight representations by applying momentum generators. The states at level $n$, built from the application of $n$ momenta on $|\mathcal{O}\rangle$, all have dimension $\Delta + n$ and states in different levels are orthogonal. Explicitly, we can label the descendants at level $N$ by a $d$-tuple $\vec{n} = \{n_1, \dots, n_d\}$ with $n_i \in \{0, \dots, N\}$ and $|\vec{n}| \equiv \sum_{i=1}^d n_i = N$. The descendant states can be expressed as

$$
|\mathcal{O}, \vec{n}\rangle = \prod_{i=1}^d P_i^{n_i} |\mathcal{O}\rangle \, .
\tag{93}
$$

Note, however, that in this basis the descendants are *not* orthogonal. The set of states that includes a primary and its descendants is called a conformal family.

These generators can also be taken to act on operators. On a primary $\mathcal{O}(x)$, we have

$$
\begin{aligned}
\left[D, \mathcal{O}(x)\right] &= (\Delta_\phi + x^\mu \partial_\mu)\mathcal{O}(x) \equiv \mathcal{D}\mathcal{O}(x), \\
\left[P_\mu, \mathcal{O}(x)\right] &= \partial_\mu \mathcal{O}(x) \equiv \mathcal{P}_\mu \mathcal{O}(x), \\
\left[K_\mu, \mathcal{O}(x)\right] &= (2x_\mu \Delta_\phi + 2x_\mu x^\nu \partial_\nu - x^2 \partial_\mu - 2x^\nu S_{\mu\nu})\mathcal{O}(x) \equiv \mathcal{K}_\mu \mathcal{O}(x), \\
\left[M_{\mu\nu}, \mathcal{O}(x)\right] &= (x_\nu \partial_\mu - x_\mu \partial_\nu + S_{\mu\nu})\mathcal{O}(x) \equiv \mathcal{M}_{\mu\nu}\mathcal{O}(x).
\end{aligned}
\tag{94}
$$

Conformal symmetry completely fixes the three-point functions of primary operators. For example, correlators of scalar primaries take the form

$$
\langle 0| \phi_1(x_1)\phi_2(x_2)\phi_3(x_3) |0\rangle = \frac{C_{\phi_1\phi_2\phi_3}}{x_{12}^{\Delta_1+\Delta_2-\Delta_3} x_{13}^{\Delta_1+\Delta_3-\Delta_2} x_{23}^{\Delta_2+\Delta_3-\Delta_1}} \, ,
\tag{95}
$$

where $C_{\phi_1\phi_2\phi_3}$ are the OPE coefficients and $x_{ij} = |x_i - x_j|$. Using this and the fact that the operator/state correspondance says that $|\mathcal{O}\rangle = \mathcal{O}(0)|0\rangle$ and $\langle\mathcal{O}| = \lim_{y\to\infty} y^{2\Delta_\mathcal{O}} \langle 0|\mathcal{O}(y)$, we find

$$
\langle\mathcal{O}| \phi(x) |\mathcal{O}\rangle = C_{\mathcal{O}\phi\mathcal{O}}(x^2)^{-\frac{\Delta_\phi}{2}} \, .
\tag{96}
$$

# B Casimir equation in general dimensions

In this section we use the orthonormal basis for the generators of the conformal algebra to derive the differential equation satisfied by the scalar-scalar conformal blocks in any dimension.

## B.1 Orthonormal basis of the conformal algebra

In this appendix, we detail the orthonormal basis for the conformal generators that is used in Section B.2. This is essentially a review of [51], although we use a different normalization for the generators. As usual when discussing rotation groups, we will have to distinguish the cases of even and odd dimension.

### B.1.1 Even dimension

We start by obtaining the basis for the rotations $M_{\mu\nu}$. In even dimensions $d = 2r$, the rotation group $SO(2r)$ has $r$ mutually commuting Cartan generators which can be simultaneously diagonalized. We will choose them to be

$$H_j = -iM_{2j-1\,2j}, \qquad j = 1, ..., r. \tag{97}$$

These are rotations in the $\{x^{2j-1}, x^{2j}\}$ planes. The rest of the generators can be organized into raising and lowering operators for the eigenvalues of the Cartans. We introduce the antisymmetric generators

$$E_{jk}^{\epsilon\eta} \equiv -iM_{2j-1\,2k-1} + \epsilon M_{2j\,2k-1} + \eta M_{2j-1\,2k} + i\eta\epsilon M_{2j\,2k}, \qquad \epsilon, \eta = \pm 1, \quad j \neq k, \tag{98}$$

which raise or lower the eigenvalue of $H_j$ and $H_k$ depending on the values of $\epsilon$ and $\eta$ respectively, as can be seen from the commutation relations

$$\begin{aligned} \left[H_i, E_{jk}^{\epsilon\eta}\right] &= (\epsilon\delta_{ij} + \eta\delta_{ik})E_{jk}^{\epsilon\eta}, \\ \left[E_{jk}^{\epsilon\eta}, E_{jk}^{\epsilon'\eta'}\right] &= (\epsilon - \epsilon')(1 - \eta\eta')H_j + (\eta - \eta')(1 - \epsilon\epsilon')H_k. \end{aligned} \tag{99}$$

Note that there are no sums over repeated indices here and we omit other commutators that are not useful to us. Note that $(E_{jk}^{\epsilon\eta})^\dagger = E_{jk}^{-\epsilon-\eta}$. The inverse of (98) can easily be found to be, for $j \neq k$,

$$\begin{aligned} M_{2j-1\,2k-1} &= \frac{i}{4}\sum_{\epsilon,\eta=\pm 1} E_{jk}^{\epsilon\eta}, & M_{2j\,2k-1} &= \frac{1}{4}\sum_{\epsilon,\eta=\pm 1} \eta E_{jk}^{\epsilon\eta}, \\ M_{2j-1\,2k} &= \frac{1}{4}\sum_{\epsilon,\eta=\pm 1} \epsilon E_{jk}^{\epsilon\eta}, & M_{2j\,2k} &= -\frac{i}{4}\sum_{\epsilon,\eta=\pm 1} \epsilon\eta E_{jk}^{\epsilon\eta}. \end{aligned} \tag{100}$$

When we consider the rest of the conformal group, we need to include an extra Cartan generator, the dilation operator $D$, which commutes with rotations. The momentum and special conformal transformation generators are organized again in terms of $\{x^{2j-1}, x^{2j}\}$ planes such that they act nicely on the Cartans. The explicit expressions are

$$P_{j\pm} \equiv P_{2j-1} \pm iP_{2j}, \qquad K_{j\pm} \equiv K_{2j-1} \pm iK_{2j}, \tag{101}$$

$$\left[H_i, P_{j\pm}\right] = \pm\delta_{ij}P_{j\pm}, \qquad \left[H_i, K_{j\pm}\right] = \pm\delta_{ij}K_{j\pm}. \tag{102}$$

These act the usual way on the eigenvalues of $D$.

### B.1.2 Odd dimension

In odd dimensions $d = 2r + 1$, the $r$ Cartan generators are the same but there are extra ladder operators. In the rotation group, we need to include the following operators

$$E_j^\pm \equiv iM_{2j\,2r+1} \pm M_{2j-1\,2r+1}, \tag{103}$$

$$\left[H_i, E_j^{\pm}\right] = \pm \delta_{ij} E_j^{\pm}, \qquad \left[E_j^{\epsilon}, E_j^{\eta}\right] = (\epsilon - \eta) H_i, \tag{104}$$

which raise or lower the eigenvalue of each Cartan separately. The inverse of this change of basis is

$$M_{2j\,2r+1} = \frac{E_j^{+} + E_j^{-}}{2i}, \qquad M_{2j-1\,2r+1} = \frac{E_j^{+} - E_j^{-}}{2}. \tag{105}$$

The extra momentum and special conformal transformation generators are just renamed, and do not act on the rotations at all since they act on a different plane:

$$P_0 \equiv P_{2r+1}, \qquad K_0 \equiv K_{2r+1}. \tag{106}$$

## B.2 Casimir equation

In this section we use the orthonormal basis for the generators of the conformal algebra discussed in Appendix B.1 to derive the differential equation satisfied by the scalar-scalar conformal blocks in any dimensions. The object that we are interested in studying is the contribution to the thermal expectation value of a scalar operator from the conformal family of another scalar. More precisely, define the projection operator

$$P_{\mathcal{O}} = \sum_{i,j=\mathcal{O},P\mathcal{O},P^2\mathcal{O},\dots} |i\rangle \, (B^{-1})_{ij} \, \langle j|, \tag{107}$$

which sums only over the states in the conformal family of $\mathcal{O}$. We can insert this into the original trace to obtain

$$H_{\Delta_{\phi},\Delta_{\mathcal{O}}}(q) = \text{Tr}\left[P_{\mathcal{O}}\phi(x)q^D\right] \equiv \text{Tr}_{\mathcal{O}}\left[\phi(x)q^D\right]. \tag{108}$$

This is, up to an OPE coefficient, the conformal block that we are looking for. To compute this object, we will use the fact that the quadratic Casimir of the conformal group has a fixed value when acting on states of a given conformal family. We can insert the Casimir operator in the trace and convert it into a differential operator acting on $H_{\Delta_{\phi},\Delta_{\mathcal{O}}}(q)$ to obtain a differential equation. It is necessary to turn on a chemical potential $y_i$ for each of the Cartan generators $H_i$ of the rotation group in order to obtain a differential equation. The result will have the form

$$\text{Tr}\left[P_{\mathcal{O}}C\phi(x)q^D \prod_{i=1}^{rank[SO(d)]} y_i^{H_i}\right] = \Delta_{\mathcal{O}}(\Delta_{\mathcal{O}} - d)H_{\Delta_{\phi},\Delta_{\mathcal{O}}}(q,\vec{y}) = \mathcal{C}H_{\Delta_{\phi},\Delta_{\mathcal{O}}}(q,\vec{y}), \tag{109}$$

with $\mathcal{C}$ the differential operator associated with the Casimir operator.

The first step in getting the differential operator is to express the conformal Casimir in terms of the orthonormal basis introduced in Appendix B.1. In general dimension, the Casimir takes the form

$$C = D(D - d) + J^2 - P_{\mu}K^{\mu},$$
$$J^2 = -\frac{1}{2}M_{\mu\nu}M^{\mu\nu}. \tag{110}$$

In odd dimensions $d = 2r + 1$, it is straightforward to get

$$P_{\mu}K^{\mu} = [P_0 K_0] + \frac{1}{2}\sum_{i=1}^{r}(P_{i+}K_{i-} + P_{i-}K_{i+}), \tag{111}$$

$$J^2 = \left[ \sum_{i=1}^{r} \left( H_i + E_j^- E_j^+ \right) \right] + \sum_{i=1}^{r} H_i^2 + \frac{1}{4} \sum_{\substack{i,j=1 \\ i \neq j}}^{r} \left( E_{ij}^{-+} E_{ij}^{+-} + E_{ij}^{--} E_{ij}^{++} \right). \tag{112}$$

In even dimensions $d = 2r$, we simply omit the extra terms in the brackets. Using this, the Casimir trick relies on the fact that we know exactly how each operator in this basis acts on the eigenvalue of the Cartans:

$$
\begin{aligned}
P_{i\pm} q^D \prod_k y_k^{H_k} &= q^{D-1} y_i^{\mp 1} \prod_k y_k^{H_k} P_{i\pm}, \quad P_0 q^D \prod_k y_k^{H_k} = q^{D-1} \prod_k y_k^{H_k} P_0, \\
K_{i\pm} q^D \prod_k y_k^{H_k} &= q^{D+1} y_i^{\mp 1} \prod_k y_k^{H_k} K_{i\pm}, \quad K_0 q^D \prod_k y_k^{H_k} = q^{D+1} \prod_k y_k^{H_k} K_0, \\
E_{ij}^{\epsilon \eta} q^D \prod_k y_k^{H_k} &= q^D y_i^{-\epsilon} y_j^{-\eta} \prod_k y_k^{H_k} E_{ij}^{\epsilon \eta}, \\
E_i^{\pm} q^D \prod_k y_k^{H_k} &= q^D y_i^{\mp 1} \prod_k y_k^{H_k} E_i^{\pm}.
\end{aligned}
\tag{113}
$$

We now have everything we need to calculate the differential operator $\mathcal{C}$. The first thing to note is that inserting $D$ in the trace $\text{Tr}_{\mathcal{O}} \left[ \phi(x) q^D \prod_i y_i^{H_i} \right]$ is equivalent to acting on it with $q \partial_q$. So we can convert the first term of the Casimir (110) into derivatives. Similarly, we can convert other terms by using the fact that inserting $H_i$ is the same as acting with $y_i \partial_{y_i}$. This is the reason why we need to include the chemical potentials in the trace. For the other operators, we use the commutation relations (94) along with (113) and the conformal algebra to bring the operators to the right of $\phi(x)$. The cyclicity of the trace allows us to combine terms and convert the quantum operators to differential operators. For example,

$$
\begin{aligned}
\text{Tr} \left[ P_0 \phi q^D \prod_k y_k^{H_k} \right] &= \text{Tr} \left[ \phi P_0 q^D \prod_k y_k^{H_k} \right] + \mathcal{P}_0 \text{Tr} \left[ \phi q^D \prod_k y_k^{H_k} \right] \\
&= q^{-1} \text{Tr} \left[ \phi q^D \prod_k y_k^{H_k} P_0 \right] + \mathcal{P}_0 \text{Tr} \left[ \phi q^D \prod_k y_k^{H_k} \right] \tag{114} \\
&\Rightarrow \text{Tr} \left[ P_0 \phi q^D \prod_k y_k^{H_k} \right] = \frac{1}{1 - q^{-1}} \mathcal{P}_0 H_{\Delta_\phi, \Delta_{\mathcal{O}}}(q, \vec{y}).
\end{aligned}
$$

The result is the Casimir equation

$$
\begin{aligned}
\Delta_{\mathcal{O}}(\Delta_{\mathcal{O}} - d) H =& \left[ (q \partial_q)^2 - d \, q \partial_q \right] H + \left[ \frac{2}{(1 - q^{-1})} (q \partial_q) H - \frac{1}{(1 - q^{-1})} \frac{1}{(1 - q)} \mathcal{P}_0 \mathcal{K}_0 H \right] \\
& \sum_{i=1}^{r} (y_i \partial_{y_i})^2 H + \left[ -\frac{1 + y_i}{1 - y_i} (y_i \partial_{y_i}) H + \frac{1}{(1 - y_i)} \frac{1}{(1 - y_i^{-1})} \mathcal{E}_i^- \mathcal{E}_i^+ H \right] \\
& + \frac{2}{1 - (q y_i)^{-1}} [(q \partial_q) + (y_i \partial_{y_i})] H - \frac{1}{2} \frac{1}{(1 - q^{-1} y_i^{-1})} \frac{1}{(1 - q y_i)} \mathcal{P}_{i+} \mathcal{K}_{i-} H \\
& + \frac{2}{1 - q^{-1} y_i} [(q \partial_q) - (y_i \partial_{y_i})] H - \frac{1}{2} \frac{1}{(1 - q^{-1} y_i)} \frac{1}{(1 - q y_i^{-1})} \mathcal{P}_{i-} \mathcal{K}_{i+} H \\
& + \sum_{\substack{j,k=1 \\ j \neq k}}^{r} \frac{1}{1 - y_j^{-1} y_k} [(y_j \partial_{y_j}) + (y_k \partial_{y_k})] H + \frac{1}{4} \frac{1}{(1 - y_j^{-1} y_k)(1 - y_j y_k^{-1})} \mathcal{E}_{jk}^{-+} \mathcal{E}_{jk}^{+-} H \\
& \frac{1}{1 - y_j y_k} [(y_j \partial_{y_j}) + (y_k \partial_{y_k})] H + \frac{1}{4} \frac{1}{(1 - y_j y_k)(1 - y_j^{-1} y_k^{-1})} \mathcal{E}_{jk}^{--} \mathcal{E}_{jk}^{++} H.
\end{aligned}
$$

In $d = 3$, for example, this equation becomes

$$
\begin{aligned}
\Delta_{\mathcal{O}}(\Delta_{\mathcal{O}} - 3)H &= \left[(q\partial_q)^2 - 3q\partial_q\right]H + (y\partial_y)^2 H \\
&\quad -\frac{1+y}{1-y}(y\partial_y)H + \frac{1}{(1-y)}\frac{1}{(1-y^{-1})}\mathcal{J}_-\mathcal{J}_+H \\
&\quad +\frac{2}{(1-q^{-1})}(q\partial_q)H - \frac{1}{(1-q^{-1})}\frac{1}{(1-q)}\mathcal{P}_0\mathcal{K}_0 H \\
&\quad +\frac{2}{1-(qy)^{-1}}[(q\partial_q)+(y\partial_y)]H - \frac{1}{2}\frac{1}{1-(qy)^{-1}}\frac{1}{(1-qy)}\mathcal{P}_+\mathcal{K}_-H \\
&\quad +\frac{2}{1-q^{-1}y}[(q\partial_q)-(y\partial_y)]H - \frac{1}{2}\frac{1}{1-q^{-1}y}\frac{1}{1-qy^{-1}}\mathcal{P}_-\mathcal{K}_+H, \quad (115)
\end{aligned}
$$

which reduces to (27) acting on $f \equiv C_{\mathcal{O}\phi\mathcal{O}}^{-1} q^{\Delta_{\mathcal{O}}}(x^2)^{\Delta_\phi/2}H$.

## C   Brute force calculation of the blocks

In this section, we will describe an algorithm to obtain coefficients in the $q$-expansion of the conformal blocks, and use it to obtain low-level coefficients. This can be implemented in any dimension; we illustrate the case of scalar blocks in $d = 3$.

The scalar blocks are defined in (8) as

$$
\mathcal{F}_{\Delta_\phi;\Delta_{\mathcal{O}}}(q) = q^{\Delta_{\mathcal{O}}} \sum_{k=0}^{\infty} q^k \sum_{|m|=|n|=k} \frac{\langle\mathcal{O},\vec{m}|\phi(1)|\mathcal{O},\vec{n}\rangle}{C_{\mathcal{O}\phi\mathcal{O}}} (B^{-1})^{\vec{m},\vec{n}} \equiv q^{\Delta_{\mathcal{O}}} \sum_{k=0}^{\infty} c_k q^k, \quad (116)
$$

where $B$ is the norm matrix at level $k$. The coefficient $c_k$'s are just numbers fixed by conformal symmetry and are defined as

$$
c_k \equiv \sum_{|m|=|n|=k} \frac{\langle\mathcal{O},\vec{m}|\phi(1)|\mathcal{O},\vec{n}\rangle}{C_{\mathcal{O}\phi\mathcal{O}}} (B^{-1})^{\vec{m},\vec{n}}. \quad (117)
$$

Note that $c_0 = 1$.

Using the conformal algebra reviewed in Appendix A, one can easily obtain the following recursion relation:

$$
\begin{aligned}
&\langle\mathcal{O}|K_3^{m_3}K_2^{m_2}K_1^{m_1}\phi P_1^{n_1}P_2^{n_2}P_3^{n_3}|\mathcal{O}\rangle \\
&= -\mathcal{P}_1 \langle\mathcal{O}|K_3^{m_3}K_2^{m_2}K_1^{m_1}\phi P_1^{n_1-1}P_2^{n_2}P_3^{n_3}|\mathcal{O}\rangle \\
&\quad +m_1(2\Delta_{\mathcal{O}}+2m_{tot}-m_1-1)\langle\mathcal{O}|K_3^{m_3}K_2^{m_2}K_1^{m_1-1}\phi P_1^{n_1-1}P_2^{n_2}P_3^{n_3}|\mathcal{O}\rangle \\
&\quad -(m_2-1)m_2\langle\mathcal{O}|K_3^{m_3}K_2^{m_2-2}K_1^{m_1+1}\phi P_1^{n_1-1}P_2^{n_2}P_3^{n_3}|\mathcal{O}\rangle \\
&\quad -(m_3-1)m_3\langle\mathcal{O}|K_3^{m_3-2}K_2^{m_2}K_1^{m_1+1}\phi P_1^{n_1-1}P_2^{n_2}P_3^{n_3}|\mathcal{O}\rangle, \quad (118)
\end{aligned}
$$

where $n_{tot} = n_1 + n_2 + n_3$ and $m_{tot} = m_1 + m_2 + m_3$. An equivalent recursion can be found for dimensions other than 3, where there would be $d$ different $m_i$'s and $n_i$'s. This relation can be implemented in Mathematica for a given level with $c_0 = 1$ as the initial condition. At low $k$,

defining $m_\phi^2 \equiv \Delta_\phi(\Delta_\phi - 3)$, the $c_k$'s are given by

$$
\begin{aligned}
c_0 &= 1, \\
c_1 &= 3 + \frac{m_\phi^2}{2\Delta_\mathcal{O}}, \\
c_2 &= 6 + \frac{m_\phi^2}{2^2(\Delta_\mathcal{O}+1)(\Delta_\mathcal{O})(2\Delta_\mathcal{O}-1)}\left[m_\phi^2\Delta_\mathcal{O} + 2\left(8\Delta_\mathcal{O}^2 + \Delta_\mathcal{O} - 2\right)\right], \\
c_3 &= 10 + \frac{m_\phi^2}{3\times 2^3(\Delta_\mathcal{O}+2)(\Delta_\mathcal{O}+1)\Delta_\mathcal{O}(2\Delta_\mathcal{O}-1)}\Big[m_\phi^4(\Delta_\mathcal{O}+1) \\
&\quad + 2m_\phi^2(15\Delta_\mathcal{O}^2 + 20\Delta_\mathcal{O} - 1) + 20(12\Delta_\mathcal{O}^3 + 21\Delta_\mathcal{O}^2 - \Delta_\mathcal{O} - 4)\Big], \\
c_4 &= 15 + \frac{m_\phi^2}{6\times 2^4\Delta_\mathcal{O}(\Delta_\mathcal{O}+1)(\Delta_\mathcal{O}+2)(\Delta_\mathcal{O}+3)(2\Delta_\mathcal{O}-1)(2\Delta_\mathcal{O}+1)} \\
&\quad \times \Big[m_\phi^6(\Delta_\mathcal{O}+1)(\Delta_\mathcal{O}+2) + 4m_\phi^4(\Delta+1)\left(12\Delta_\mathcal{O}^2 + 31\Delta_\mathcal{O} + 8\right) \\
&\quad + 4m_\phi^2\left(180\Delta_\mathcal{O}^4 + 750\Delta_\mathcal{O}^3 + 829\Delta_\mathcal{O}^2 + 201\Delta_\mathcal{O} - 22\right) \\
&\quad + 240\left(16\Delta_\mathcal{O}^5 + 78\Delta_\mathcal{O}^4 + 105\Delta_\mathcal{O}^3 + 26\Delta_\mathcal{O}^2 - 17\Delta_\mathcal{O} - 6\right)\Big].
\end{aligned}
\tag{119}
$$

A simple consistency check is that in when $m_\phi = 0$ this reduces to the character

$$
(1-q)^{-3} = 1 + 3q + 6q^2 + 10q^3 + 15q^4 + 21q^5 + \dots
\tag{120}
$$

counting the number of states at each level.

## D  Various limits of the 3D conformal block using Casimir equations

The differential equation Eq. (27) is difficult to study in general, so we now describe two limits which are easier to study. This will give us a slight generalization of the results of Secs. 3.3.1 and 3.3.2.

### D.1  WKB Limit

We first consider the WKB limit. Let $\Delta_\phi \to \infty$ and $\Delta_\mathcal{O} \to \infty$ with fixed $\rho = \Delta_\mathcal{O}/\Delta_\phi$. Inserting the ansatz $f = e^{-\Delta_\phi G}$ into the differential equation we obtain the leading equation

$$
\begin{aligned}
0 &= q^2(\partial_q G)^2 + u(u+4)(\partial_u G)^2 - 2q\rho^{-1}(\partial_q G) - 4u^{-1}s(s-1)(\partial_s G)^2 \\
&\quad + (s-1)\frac{q}{(1-q)^2}\left[1 - 4s(\partial_s G) + 4s^2(\partial_s G)^2\right] \\
&\quad + \left[\frac{q(q(q(u+2)-4)+u+2)}{2(q^2 - q(u+2)+1)^2}\right]\left[-s + 4s(s-1)(\partial_s G) - 4s(s-1)^2(\partial_s G)^2\right].
\end{aligned}
\tag{121}
$$

Now we expand $G = \sum G_k u^k$ to obtain (the leading equation implies that $\partial_s G_0 = 0$):

$$
0 = q(\partial_q G_0)^2 - 2\rho^{-1}(\partial_q G_0) - \frac{1}{(1-q)^2} \quad \Rightarrow \quad \partial_q G_0(q) = \frac{1}{q\rho} + \frac{\sqrt{q^2 + (\rho^2 - 2)q + 1}}{(q-1)q\rho},
\tag{122}
$$

where we have picked a particular sign in solving the quadratic equation to ensure that there is no $\log q$ term in $G_0(q)$. This agrees with Eq. (70).

### D.2 Large $\Delta_{\mathcal{O}}$ limit

We now consider the differential equation (27) in the limit where $\Delta_{\mathcal{O}} \to \infty$ while $\Delta_{\phi}$ is fixed. Most of the terms in the equation are subleading in this limit, and we get

$$0 = \partial_q f + \frac{(3q^2 - 2q(u+3) + u + 3)}{(q-1)^3 - (q-1)qu} f. \tag{123}$$

The solution is

$$f(q,u,s) = \frac{1}{(1-q)^3 - (1-q)qu}, \tag{124}$$

where we have used the $q \to 0$ limit to fix the boundary condition. This reproduces the result of Section 3.3.2 when $u = 0$, in the case $d = 3$.

To calculate the first correction in the $1/\Delta_{\mathcal{O}}$ expansion, we need to expand in the $s$ and $u$ variables as

$$f(q,u,s) = \frac{1}{(1-q)^3 - (1-q)qu} + \sum_{a=1}^{\infty} \sum_{b,c=0}^{\infty} f_{b,c}^{(a)}(q) \Delta_{\mathcal{O}}^{-a} u^b s^c. \tag{125}$$

Since we care about the usual blocks obtained by setting $u = 0$, we only want to calculate $f_{0,0}^{(1)}(q)$. Using this expansion in (27) and asking that $f_{0,0}^{(1)}(0) = 0$ (the contribution from the primary doesn't depend on $\Delta_{\mathcal{O}}$), we quickly find that

$$f_{0,0}^{(1)}(q) = \frac{\Delta_{\phi}(\Delta_{\phi} - 3)}{2} \frac{q}{(1-q)^4}. \tag{126}$$

Again this agrees with Section 3.3.2.

## E  AdS-integral representation satisfies the Casimir differential equation

In this section we will prove the AdS integral representation of the conformal block for thermal one point functions by showing that it satisfies the Casimir equation derived in Sec. 2 for $d = 3$, and in Appendix B for general dimension. The fact that the AdS-integral obeys the correct boundary condition follows by the same arguments as in Sec. 3.

### E.1  Field theory considerations

As in the previous sections, the Casimir equation can be derived by inserting a Casimir operator into the thermal block. On the one hand, the Casimir operator gives the same value for all the states in a representation; on the other hand, by Ward identities, the insertion of an operator can be transformed into a set of derivatives on the conformal block. In this subsection we will derive a recursion relation for this operator, which will be related to Witten diagrams in the next subsection.

In a CFT$_d$, we can separate the conformal algebra into two sets of operators, $H_a$ and $S_i$. $H_a$ is the Cartan subalgebra, including the dilation operator and the Cartan subalgebra of the rotation group $SO(d)$. Comparing with Appendix B.2, we call the dilation operator $H_0$ in

order to simplify our notation. The remaining operators $S_i$ are chosen to satisfy the eigenvalue equation (no sum over repeated indices)

$$[H_a, S_i] = w_{i,a} S_i. \tag{127}$$

The $w_{i,a}$ are the roots, which tell us how the $S_i$ change the $H_a$ eigenvalues of a state. We wish to calculate

$$\mathcal{F} = \mathrm{Tr}_{\mathcal{O}}\left[\phi(x)\prod_{a=0}^{r} y_a^{H_a}\right], \tag{128}$$

where the trace is only over the conformal family of the scalar $\mathcal{O}$ and $y_0$ is the variable $q$ used earlier. The number $r$ is the rank of the conformal algebra. As in Section 2, the conformal block for the thermal expectation value of the scalar $\phi(x)$ is given by the limit $y_a \to 1$ for $a \neq 0$. In the rest of this section we will suppress the product symbol for simplicity, so anything of the form $y_a^{A_a}$ implies a product over $a$.

Now we are ready to define the quantity

$$\mathcal{F}(V) \equiv \mathrm{Tr}_{\mathcal{O}}\left[V\phi(x)y_b^{H_b}\right], \tag{129}$$

which is the trace with insertion of an operator $V$. In what follows, we will derive a recursion relation which transforms an insertion of $V$ into derivatives acting on the conformal block. The first important relation is simply

$$\mathcal{F}(H_a V) = \mathrm{Tr}_{\mathcal{O}}\left[H_a V\phi(x)y_b^{H_b}\right] = y_a\frac{\partial}{\partial y_a}\mathrm{Tr}_{\mathcal{O}}\left[V\phi(x)y_b^{H_b}\right] = y_a\frac{\partial}{\partial y_a}\mathcal{F}(V). \tag{130}$$

The second relation is obtained by using the roots to pass $S_i$ through $H_a$:

$$
\begin{aligned}
\mathcal{F}(S_i V) &= \mathrm{Tr}_{\mathcal{O}}\left[y_b^{H_b} S_i V\phi(x)\right] = y_a^{w_{i,a}}\mathrm{Tr}_{\mathcal{O}}\left[S_i y_b^{H_b} V\phi(x)\right] = y_a^{w_{i,a}}\mathrm{Tr}_{\mathcal{O}}\left[y_b^{H_b} V\phi(x)S_i\right] \\
&= y_a^{w_{i,a}}\mathrm{Tr}_{\mathcal{O}}\left[y_b^{H_b} V S_i\phi(x)\right] + y_a^{w_{i,a}}\mathrm{Tr}_{\mathcal{O}}\left[y_b^{H_b} V[\phi(x), S_i]\right] \\
&= y_a^{w_{i,a}}(\mathcal{F}(S_i V) - \mathcal{F}([S_i, V])) - y_a^{w_{i,a}}\mathcal{S}_i\mathcal{F},
\end{aligned}
\tag{131}
$$

so we have

$$(1 - y_a^{w_{i,a}})\mathcal{F}(S_i V) = -y_a^{w_{i,a}}\mathcal{F}([S_i, V]) - y_a^{w_{i,a}}\mathcal{S}_i\mathcal{F}. \tag{132}$$

Using these relations recursively, we can easily transform the insertion of Casimir operator into a second order derivative on the conformal block and derive the Casimir equation.

## E.2 Solution to Casimir differential equation

In this section, we will show that the Witten diagram in global $\mathrm{AdS}_{d+1}$ obeys a similar set of recursion relations. In this subsection, the bulk field dual to the boundary scalar $\phi$ will be denoted $\hat{\phi}$, $x$ denotes a boundary point, and $y$ denotes a bulk point. The internal bulk scalar that runs in the loop and is dual to $\mathcal{O}$ will be called $\hat{\mathcal{O}}$.

The propagators can be written as scalar two-point functions in $\mathrm{AdS}_{d+1}$ space. The bulk to bulk propagator between two bulk points $y$ and $y'$ is

$$G_{bb}^{\Delta_{\mathcal{O}}}(y, y') = \langle\hat{\mathcal{O}}(y)\hat{\mathcal{O}}(y')\rangle. \tag{133}$$

We can then obtain the bulk to boundary propagator by taking one bulk field to the boundary and removing the scaling factor.

The crucial point is that the isometries of $\text{AdS}_{d+1}$ are the conformal transformations of a $\text{CFT}_d$. The generators of these isometries (using a hat to distinguish them from the analogous CFT operators) act on a bulk scalar field as

$$[\hat{L}, \Phi(y)] = \hat{\mathcal{L}}\Phi(y) \tag{134}$$

for some differential operator $\hat{\mathcal{L}}$. The (bulk) Casimir operator $\hat{\mathcal{C}}$ of the conformal group can be expressed in terms of these differential operators in the same way as in the CFT. The bulk-to-bulk propagator satisfies by definition the equation

$$\hat{\mathcal{C}}_y G_{BB}^{(\mathcal{O})}(y, y') = \Delta_{\mathcal{O}}(\Delta_{\mathcal{O}} - d) G_{BB}^{(\mathcal{O})}(y, y'). \tag{135}$$

The left hand side of this equation can be rewritten as

$$\left\langle \hat{\mathcal{O}}(y')[\hat{C}, \hat{\mathcal{O}}(y)]_{\text{Adj}} \right\rangle = \left\langle \hat{\mathcal{O}}(y')\hat{C}\hat{\mathcal{O}}(y) \right\rangle, \tag{136}$$

where the subscript Adj indicates that we are acting on the operator in the adjoint representation. As the vacuum is invariant under the isometries, the terms with isometry operators on the right hand side vanish. We then just need to convert the quantum operators into derivatives, just as in our CFT discussion. To do so, we will need a few relations for the propagator which we derive below.

To begin, let us write our proposed integral formula for the one point conformal block as

$$\hat{\mathcal{F}} = \int dy^{d+1} \sqrt{g(y)} \langle e^{-i\lambda^a \hat{H}_a} \hat{\mathcal{O}}(y) e^{i\lambda^b \hat{H}_b} \hat{\mathcal{O}}(y) \rangle G_{B\partial}^{(\phi)}(y, x). \tag{137}$$

Here $y_a \equiv e^{i\lambda^a}$ are the thermodynamic potentials for the Cartans $H_a$; as above the product over $a$ is implied. The Killing generators annihilate the vacuum, so we can ignore the $e^{-i\lambda^a \hat{H}_a}$ term. We further define

$$\hat{\mathcal{F}}(\hat{V}) = \int dy^{d+1} \sqrt{g(y)} \langle \hat{\mathcal{O}}(y) y_b^{\hat{H}_b} \hat{V} \hat{\mathcal{O}}(y) \rangle G_{B\partial}^{(\hat{\mathcal{O}})}(y, x), \tag{138}$$

which allows us to obtain the desired recursion relations. The first one is simply

$$\hat{\mathcal{F}}(\hat{H}_a \hat{V}) = \int dy^{d+1} \sqrt{g(y)} \langle \hat{\mathcal{O}}(y) y_b^{\hat{H}_b} \hat{H}_a \hat{V} \hat{\mathcal{O}}(y) \rangle G_{B\partial}^{(\phi)}(y, x) = y_a \frac{\partial}{\partial y_a} \hat{\mathcal{F}}(\hat{V}). \tag{139}$$

We can also insert the $\hat{S}_i$ operators and use the roots of the conformal algebra along with the fact that $\hat{S}_i$ kills the vacuum to write

$$
\begin{aligned}
\left\langle \hat{\mathcal{O}}(y) y_b^{\hat{H}_b} \hat{S}_i \hat{V} \hat{\mathcal{O}}(y) \right\rangle &= y_a^{w_{i,a}} \left\langle \hat{\mathcal{O}}(y) \hat{S}_i y_b^{\hat{H}_b} \hat{V} \hat{\mathcal{O}}(y) \right\rangle = y_a^{w_{i,a}} \left\langle [\hat{\mathcal{O}}(y), \hat{S}_i] y_b^{\hat{H}_b} \hat{V} \hat{\mathcal{O}}(y) \right\rangle \\
&= -y_a^{w_{i,a}} \left\langle (\hat{S}_i \hat{\mathcal{O}}(y)) y_b^{\hat{H}_b} \hat{V} \hat{\mathcal{O}}(y) \right\rangle \\
&= -y_a^{w_{i,a}} \hat{S}_i \left\langle \hat{\mathcal{O}}(y) y_b^{\hat{H}_b} \hat{V} \hat{\mathcal{O}}(y) \right\rangle + y_a^{w_{i,a}} \left\langle \hat{\mathcal{O}}(y) y_b^{\hat{H}_b} \hat{V} (\hat{S}_i \hat{\mathcal{O}}(y)) \right\rangle \\
&= -y_a^{w_{i,a}} \hat{S}_i \left\langle \hat{\mathcal{O}}(y) y_b^{\hat{H}_b} \hat{V} \hat{\mathcal{O}}(y) \right\rangle + y_a^{w_{i,a}} \left\langle \hat{\mathcal{O}}(y) y_b^{\hat{H}_b} \hat{V} [\hat{S}_i, \hat{\mathcal{O}}(y)] \right\rangle \\
&= -y_a^{w_{i,a}} \hat{S}_i \left\langle \hat{\mathcal{O}}(y) y_b^{\hat{H}_b} \hat{V} \hat{\mathcal{O}}(y) \right\rangle + y_a^{w_{i,a}} \left\langle \hat{\mathcal{O}}(y) y_b^{\hat{H}_b} \hat{V} \hat{S}_i \hat{\mathcal{O}}(y) \right\rangle.
\end{aligned} \tag{140}
$$

The first term of this expression can be integrated by parts to move the differential operator to the bulk to boundary propagator

$$
\begin{aligned}
\hat{\mathcal{F}}(\hat{S}_i\hat{V}) &= \int dy^{d+1}\sqrt{g(y)}\left\langle \hat{\mathcal{O}}(y)y_b^{\hat{H}_b}\hat{S}_i\hat{V}\hat{\mathcal{O}}(y)\right\rangle G_{B\partial}^{(\phi)}(y,x) \\
&= y_a^{w_{a,i}}\int dy^{d+1}\sqrt{g(y)}\left\langle \hat{\mathcal{O}}(y)y_b^{\hat{H}_b}\hat{V}\hat{\mathcal{O}}(y)\right\rangle \hat{\mathcal{S}}_i^{(y)}G_{B\partial}^{(\phi)}(y,x) \\
&\quad + y_a^{w_{i,a}}\int dy^{d+1}\sqrt{g(y)}\left\langle \hat{\mathcal{O}}(y)y_b^{\hat{H}_b}\hat{V}\hat{S}_i\hat{\mathcal{O}}(y)\right\rangle G_{B\partial}^{(\phi)}(y,x) \\
&= -y_a^{w_{i,a}}\int dy^{d+1}\sqrt{g(y)}\left\langle \hat{\mathcal{O}}(y)y_b^{\hat{H}_b}\hat{V}\hat{\mathcal{O}}(y)\right\rangle \mathcal{S}_i^{(x)}G_{B\partial}^{(\phi)}(y,x) \\
&\quad + y_a^{w_{i,a}}\int dy^{d+1}\sqrt{g(y)}\left\langle \hat{\mathcal{O}}(y)y_b^{\hat{H}_b}\hat{V}\hat{S}_i\hat{\mathcal{O}}(y)\right\rangle G_{B\partial}^{(\phi)}(y,x) \\
&= -y_a^{w_{i,a}}\mathcal{S}_i\hat{\mathcal{F}}(\hat{V}) + y_a^{w_{i,a}}\hat{\mathcal{F}}(\hat{V}\hat{S}_i)\,.
\end{aligned}
\tag{141}
$$

So

$$
(1-y_a^{w_{i,a}})\hat{\mathcal{F}}(\hat{S}_i\hat{V}) = -y_a^{w_{i,a}}\mathcal{S}_i\hat{\mathcal{F}}(\hat{V}) - y_a^{w_{i,a}}\hat{\mathcal{F}}([\hat{S}_i,\hat{V}]).
\tag{142}
$$

This is exactly the same recursion relation derived for the boundary differential operators in the CFT. The only subtlety here is that we must convert the bulk differential operator $\hat{S}_i$ into a boundary operator $\mathcal{S}_i$ by taking the bulk field to the boundary.

As in the CFT case, these relations can be used to find a differential equation obeyed by the block. Because the recursion relations are the same, the differential equation will also be the same. If we impose the same boundary conditions, this then implies that our bulk integral $\hat{\mathcal{F}}$ will then equal the conformal block $\mathcal{F}$. To check the boundary condition, we just need to investigate the low temperature behavior ($q \to 0$). In this limit, the bulk-to-bulk propagator simplifies, since we are computing the propagator between a point and its thermally-translated image. As $q \to 0$, the geodesic distance between these points goes to $e^{-\Delta_{\mathcal{O}}\beta} = q^{\Delta_{\mathcal{O}}}$, which is the correct behavior.

## F   Details of performing AdS integrals

In this section, we fill in some of the details of the calculation of the AdS integral in Sec. 3.3. The first integral to be performed is Eq. (62), which can be carried out as follows:

$$
\begin{aligned}
J &\equiv \int_{-\infty}^{\infty} dt\, d^{d-1}\Omega\left[\sqrt{1+r^2}\cosh(t-t_\infty) - r\cos\Theta(\Omega,\Omega_\infty)\right]^{-\Delta_\phi} \\
&= \frac{2\pi^{\frac{d-1}{2}}}{\Gamma\left(\frac{d-1}{2}\right)}\int_{-\infty}^{\infty}dt\int_0^\pi d\Theta\,\sin^{d-2}\Theta\left[\sqrt{1+r^2}\cosh t - r\cos\Theta\right]^{-\Delta_\phi} \\
&= \frac{2\pi^{\frac{d-1}{2}}}{\Gamma\left(\frac{d-1}{2}\right)}\int dt\, d\Theta\,\sin^{d-2}\Theta\left[\sqrt{1+r^2}\cosh t\right]^{-\Delta_\phi}\left[1 - \frac{r}{\sqrt{1+r^2}\cosh t}\cos\Theta\right]^{-\Delta_\phi} \\
&= \frac{2\pi^{\frac{d}{2}}}{\Gamma\left(\frac{d}{2}\right)}\int dt\left[\sqrt{1+r^2}\cosh t\right]^{-\Delta_\phi}{}_2F_1\left[\frac{1}{2}\Delta_\phi,\frac{1}{2}\Delta_\phi+\frac{1}{2},\frac{d}{2};\frac{r^2}{(1+r^2)\cosh^2 t}\right] \\
&= \frac{2\pi^{\frac{(d+1)}{2}}}{\Gamma\left(\frac{d}{2}\right)}\sum_{n=0}^\infty \frac{\left(\frac{\Delta_\phi}{2}\right)_n\left(\frac{\Delta_\phi}{2}+\frac{1}{2}\right)_n}{(n!)(\frac{d}{2})_n}\frac{\Gamma\left(n+\frac{\Delta_\phi}{2}\right)}{\Gamma\left(n+\frac{\Delta_\phi}{2}+\frac{1}{2}\right)}\times r^{2n}\left[1+r^2\right]^{-n-\frac{1}{2}\Delta_\phi}
\end{aligned}
$$

$$= \frac{2\pi^{\frac{(d+1)}{2}}}{\Gamma\left(\frac{d}{2}\right)} \frac{\Gamma\left(\frac{\Delta_\phi}{2}\right)}{\Gamma\left(\frac{1}{2}\left(\Delta_\phi+1\right)\right)} \left[1+r^2\right]^{-\frac{1}{2}\Delta_\phi} {}_2F_1\left(\frac{\Delta_\phi}{2}, \frac{\Delta_\phi}{2}; \frac{d}{2}; \frac{r^2}{r^2+1}\right). \tag{143}$$

In the first step, we performed the $\Theta$-integral using

$$\int_0^\pi d\Theta \ [1+c\cos\Theta]^{-b} \sin^{d-2}\Theta = \int_{-1}^{1} dx \ [1+cx]^{-b}\left(1-x^2\right)^{\frac{1}{2}(d-3)}$$
$$= \frac{\sqrt{\pi}\Gamma\left(\frac{d-1}{2}\right)}{\Gamma\left(\frac{d}{2}\right)} {}_2F_1\left[\frac{1}{2}b, \frac{1}{2}b+\frac{1}{2}, \frac{d}{2}; c^2\right]. \tag{144}$$

For this integral to converge we need $|c| \leq 1$, which is true in our case since $c = \frac{r}{\sqrt{1+r^2}\cosh t}$, and $d > 1$. In the next step, we again used the fact that $\frac{r^2}{(1+r^2)}\frac{1}{\cosh^2 t} \leq 1$, so we are allowed to use the series expansion of the hypergeometric function. Then, we performed the integral over $t$ using

$$\int_{-\infty}^{\infty} dt \cosh^{-\alpha} t = 2\int_0^\infty dt \cosh^{-\alpha} t = 2\int_1^\infty dx(x^2-1)^{-\frac{1}{2}}x^{-\alpha} = \frac{\sqrt{\pi}\Gamma\left(\frac{\alpha}{2}\right)}{\Gamma\left(\frac{\alpha+1}{2}\right)} \quad ; \quad \alpha > 0. \tag{145}$$

The requirement for the convergence of the integral is $\Delta_\phi > 0$. Now we multiply $J$ by the volume factor $r^{d-1}$ together with the bulk-to-bulk propagator evaluated at $X$ and $X_\beta$, and obtain:

$$\begin{aligned} I =& \ \mathcal{C}_{\Delta_\mathcal{O}}\mathcal{C}_{\Delta_\phi} 2^{-\Delta_\phi} \frac{2\pi^{\frac{(d+1)}{2}}}{\Gamma\left(\frac{d}{2}\right)} \frac{\Gamma\left(\frac{\Delta_\phi}{2}\right)}{\Gamma\left(\frac{1}{2}(\Delta_\phi+1)\right)} q^{\Delta_\mathcal{O}}(1-q)^{-2\Delta_\mathcal{O}} \int_0^\infty dr \ r^{d-1}\left(1+r^2\right)^{-\Delta_\mathcal{O}-\frac{1}{2}\Delta_\phi} \\ & \times {}_2F_1\left(\frac{\Delta_\phi}{2}, \frac{\Delta_\phi}{2}; \frac{d}{2}; \frac{r^2}{r^2+1}\right) {}_2F_1\left[\Delta_\mathcal{O}, \frac{2\Delta_\mathcal{O}-d+1}{2}; 2\Delta_\mathcal{O}-d+1; -\frac{4q}{(1-q)^2(1+r^2)}\right] \\ =& \ \mathcal{C}_{\Delta_\mathcal{O}}\mathcal{C}_{\Delta_\phi} 2^{-\Delta_\phi} \frac{2\pi^{\frac{(d+1)}{2}}}{\Gamma\left(\frac{d}{2}\right)} \frac{\Gamma\left(\frac{\Delta_\phi}{2}\right)}{\Gamma\left(\frac{1}{2}(\Delta_\phi+1)\right)} \frac{\Gamma\left(\frac{d}{2}\right)\Gamma\left(\Delta_\mathcal{O}-\frac{\Delta_\phi}{2}\right)\Gamma\left(\frac{1}{2}(\Delta_\phi-d)+\Delta_\mathcal{O}\right)}{2\Gamma(\Delta_\mathcal{O})^2} q^{\Delta_\mathcal{O}}(1-q)^{-2\Delta_\mathcal{O}} \\ & \times {}_3F_2\left(-\frac{d}{2}+\Delta_\mathcal{O}+\frac{1}{2}, \Delta_\mathcal{O}-\frac{\Delta_\phi}{2}, -\frac{d}{2}+\frac{\Delta_\phi}{2}+\Delta_\mathcal{O}; \Delta_\mathcal{O}, -d+2\Delta_\mathcal{O}+1; -\frac{4q}{(q-1)^2}\right). \end{aligned} \tag{146}$$

In deriving the final line of Eq. (146), we have expanded the two hypergeometric functions as a series expansion of the argument. Then using

$$\int_0^\infty r^{d-1+2n}(1+r^2)^{-\frac{1}{2}\Delta_\phi-\Delta_\mathcal{O}-n-k} = \frac{\Gamma\left(\frac{d}{2}+n\right)\Gamma\left(-\frac{d}{2}+k+\frac{\Delta_\phi}{2}+\Delta_\mathcal{O}\right)}{2\Gamma\left(k+n+\frac{\Delta_\phi}{2}+\Delta_\mathcal{O}\right)} \tag{147}$$

and performing the double sum, up to the overall normalization, one obtains the final line. This integral can be used only for $\Delta_\phi + 2\Delta > d$ and the resummation can be performed only when $\Delta > \frac{\Delta_\phi}{2}$. Together, these conditions can be combined to give $\Delta > \frac{d}{4}$.

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
