# Peer review of "Thermal Conformal Blocks"

_SciPost Physics, doi:SciPost Phys. 7, 015 (2019)_

## Round 2 · Referee Report · Anonymous (Referee 1) · 2019-5-31

Strengths

1- Explores a new idea in subject of current interest, namely using conformal bootstrap for thermal correlators.

2- Interesting and detailed results for the thermal conformal blocks

3- Well written with a good explanation of the main ideas and results.

Report

Very interesting paper. Thermal conformal blocks are not much explored but certainly deserve more interest. The authors wrote a very good paper that should be a basic future reference on this subject.

---

## Round 2 · Referee Report · Anonymous (Referee 2) · 2019-7-8

Strengths

  1. The authors address a foundational problem in conformal field theory.

  2. Their analysis is detailed, careful and, for the observable they chose to compute, complete.

  3. The combination of field theoretic and holographic techniques gives further confidence in the veracity of the result, and suggests how one might extend their result to other related blocks.

  4. The paper is clearly written without much fluff.

Weaknesses

There is nothing really worth mentioning.

Report

This paper addresses a question which will surely be valuable as we become more adept at conformal field theory above the ground state: what are the conformal blocks for a CFT on the sphere at finite temperature? We are all familiar with the works of Dolan and Osborn which established the modern basics of conformal blocks in the vacuum, and I expect this paper to have a similar status in the world of finite temperature.

The authors restricted attention to scalar blocks, as well they should, but an obvious extension of this work that would be needed in any application to a bona fide CFT would be to include operator spin. This paper lays groundwork for that.

The blocks they find are rather elegant, at least by current bootstrap standards, being generalized hypergeometric functions. The existence of a closed-form expression is itself useful.

The application to light-heavy-heavy OPE asymptotics was a nice touch, that is more rigorous and easier to understand than other similar ETH-driven approaches to these coefficients.

In conclusion, while the paper does not address the world's most adventurous topic, it is sure to be an important technical result going forward and did provide some nice physical holographic interpretation of the result.

---

## Editorial Decision

published